# Global atlas of predicted functional domains in *Legionella pneumophila* Dot/Icm translocated effectors

Deepak T Patel [1], Peter J Stogios[2], Lukasz Jaroszewski[3], Malene L Urbanus [4], Mayya Sedova [3], Cameron Semper[1], Cathy Le [1], Abraham Takkouche[3], Keita Ichii[3], Julie Innabi [3], Dhruvin H Patel [1], Alexander W Ensminger [4,5 ✉], Adam Godzik [3 ✉] & Alexei Savchenko [1,2 ✉]

## Abstract

***Legionella pneumophila* utilizes the Dot/Icm type IVB secretion system to deliver hundreds of effector proteins inside eukaryotic cells to ensure intracellular replication. Our understanding of the molecular functions of the largest pathogenic arsenal known to the bacterial world remains incomplete. By leveraging advancements in 3D protein structure prediction, we provide a comprehensive structural analysis of 368 *L. pneumophila* effectors, representing a global atlas of predicted functional domains summarized in a database (https://pathogens3d.org/legionella-pneumophila). Our analysis identified 157 types of diverse functional domains in 287 effectors, including 159 effectors with no prior functional annotations. Furthermore, we identified 35 cryptic domains in 30 effector models that have no similarity with experimentally structurally characterized proteins, thus, hinting at novel functionalities. Using this analysis, we demonstrate the activity of thirteen functional domains, including three cryptic domains, predicted in *L. pneumophila* effectors to cause growth defects in the *Saccharomyces cerevisiae* model system. This illustrates an emerging strategy of exploring synergies between predictions and targeted experimental approaches in elucidating novel effector activities involved in infection.**

**Keywords** Bacterial Effectors; *Legionella pneumophila*; Protein Modeling; Yeast Toxicity; Cryptic Domains
**Subject Categories** Computational Biology; Microbiology, Virology & Host Pathogen Interaction; Structural Biology

## Introduction

The Gram-negative bacterium, *Legionella pneumophila*, is an intracellular pathogen of freshwater protozoa (Abu Kwaik et al, 1998). The ubiquitous presence of this bacterium in human-made and natural freshwater reservoirs often leads to the accidental infection of humans from the inhalation of contaminated aerosolized water particles (Blatt et al, 1993; Muder et al, 1986). This can lead to a severe, life-threatening form of pneumonia, called Legionnaires' disease, or a self-resolving, flu-like illness, known as Pontiac fever (Cordes and Fraser, 1980; Cunha et al, 2016).

Upon being phagocytosed by the eukaryotic host cell, *L. pneumophila* remodels the phagosome into a replication-permissive compartment - termed the *Legionella*-containing vacuole (LCV) (Horwitz and Maxfield, 1984; Roy et al, 1998; Tilney et al, 2001). The establishment of the LCV is dependent on the delivery of specific proteins, called "effectors", into the host cell, which is mediated by the Dot/Icm (<u>d</u>efective in <u>o</u>rganelle <u>t</u>rafficking/<u>i</u>ntra<u>c</u>ellular <u>m</u>ultiplication) type IVB secretion system (T4SS)—an essential molecular syringe-like complex that is conserved in all species of *Legionella* (Berger and Isberg, 1993; Ensminger and Isberg, 2009; Marra et al, 1992; Ninio and Roy, 2007; Segal and Shuman, 1997). The Dot/Icm effectors are involved in the manipulation of a wide variety of host cellular processes, including vesicle trafficking, protein translation, autophagy, vacuolar function, and the cytoskeleton to avoid lysosomal fusion and for the formation of the LCV (Horwitz and Maxfield, 1984; Lockwood et al, 2022; Mondino et al, 2020b; Shames, 2023; Swanson and Isberg, 1995; Tilney et al, 2001).

Over 360 Dot/Icm-translocated effectors have been identified in *L. pneumophila* through a variety of methods, including large-scale experimental screens (Huang et al, 2011; Zhu et al, 2011) and machine-learning approaches (Burstein et al, 2009). This represents the largest arsenal of bacterial effectors described to date, with effectors representing over 10% of the *L. pneumophila* proteome. Across the entire *Legionella* genus, the number of effectors is

[1]Department of Microbiology, Immunology and Infectious Diseases, University of Calgary, Calgary, AB T2N 4N1, Canada. [2]BioZone, Department of Chemical Engineering and Applied Chemistry, University of Toronto, Toronto, ON M5S 1A4, Canada. [3]University of California, Riverside, School of Medicine, Biosciences Division, Riverside, CA, USA. [4]Department of Biochemistry, University of Toronto, Toronto, ON M5G 1M1, Canada. [5]Department of Molecular Genetics, University of Toronto, Toronto, ON M5G 1M1, Canada. ✉E-mail: alex.ensminger@utoronto.ca; adam.godzik@medsch.ucr.edu; alexei.savchenko@ucalgary.ca

staggering, with over 18,000 unique effector-coding sequences (Gomez-Valero et al, 2019). This extensive arsenal of host-manipulating factors in *Legionella* species is attributed to the rapid evolution necessary for the successful survival and colonization of diverse protozoan species in the natural habitat of *Legionella* (Amaro et al, 2015; Gomez-Valero and Buchrieser, 2019; O'Connor et al, 2011; O'Connor et al, 2012; Park et al, 2020). The ability of *L. pneumophila* and other *Legionella* species to infect human macrophages suggests that the effector arsenal targets conserved eukaryotic cellular processes. This hypothesis underscores how understanding the functions of individual *L. pneumophila* effectors could illuminate basic eukaryotic cellular processes and evolutionarily conserved pathways required for intracellular bacterial pathogenesis. The size of this arsenal, however, presents its own experimental challenges: significant functional redundancy has been observed between effectors (O'Connor et al, 2011; O'Connor et al, 2012; Park et al, 2020), limiting the effectiveness of traditional forward genetic approaches to defining effector function. Consequently, a significant number of *L. pneumophila* effectors remain functionally uncharacterized (Lockwood et al, 2022; Mondino et al, 2020b; O'Connor et al, 2012).

Insights into the molecular functions of *L. pneumophila* effectors have resulted primarily from primary sequence analysis and the detection of motifs and domains associated with known activities (Burstein et al, 2016; Gomez-Valero et al, 2019; Nachmias et al, 2019). A global evaluation of primary sequences revealed the prevalence of eukaryotic-like motifs/domains in *L. pneumophila* effectors, defined as predominantly (more than 75%) found in eukaryotic species (Gomez-Valero et al, 2019). This observation led to the hypothesis that *Legionella* acquired genes through horizontal gene transfer during co-evolution with its eukaryotic hosts, co-opting eukaryotic genes as effectors for host manipulation (Cazalet et al, 2004; de Felipe et al, 2005; Lurie-Weinberger et al, 2010). Consequently, the presence of eukaryotic-like domains or sequence similarity (over 20%) to eukaryotic proteins in *Legionella* has been used to predict effector function (Gomez-Valero et al, 2019). This predictive method proved particularly successful for effectors sharing significant similarity with functional domains associated with eukaryotic-specific processes, such as protein ubiquitination and phosphorylation (Bruckert and Abu Kwaik, 2016; Ensminger and Isberg, 2010; Kubori et al, 2008; Lee et al, 2020; Lee and Machner, 2018; Michard et al, 2015; Moss et al, 2019; Qiu and Luo, 2017; Quaile et al, 2015).

However, functional annotation of *Legionella* effectors based on their primary sequence analysis has limitations since a large proportion of these proteins do not share significant amino acid similarity with functionally characterized proteins (Gomez-Valero and Buchrieser, 2019; Mondino et al, 2020a). Consequently, experimental determination of effector three-dimensional (3D) structures—primarily using X-ray crystallography, nuclear magnetic resonance spectroscopy, and, as of recently, cryogenic electron microscopy—proved to be instrumental in revealing otherwise cryptic domains and other molecular features indicative of their activity. For example, the structure of Lpg2511/SidC revealed the presence of an N-terminal domain with a unique fold, featuring a conserved amino acid arrangement similar to the Cys-His-Asp catalytic triad found in cysteine proteases (Hsu et al, 2014). Subsequent biochemical analysis showed that the catalytic triad in Lpg2511/SidC is involved in ubiquitin ligase activity via a non-

canonical catalytic mechanism distinct from eukaryotic E3 enzymes (Hsu et al, 2014). Likewise, the crystal structure of Lpg2147/MavC revealed a domain similar in structure to ubiquitin-like protein-specific deaminase domains found in effectors of other bacterial pathogens (Samba-Louaka et al, 2009; Valleau et al, 2018). This discovery facilitated the demonstration that Lpg2147/MavC can deaminate human ubiquitin at Gln40 (Gan et al, 2020; Valleau et al, 2018) and catalyze the transglutamination of Ub via its Gln40 to Lys92 of the human UBE2N ubiquitin-conjugating enzyme (Gan et al, 2020; Mu et al, 2020). These examples underscore the advantages of molecular structural information for characterizing the activity of an effector in the host cell. Yet, the experimental determination of effector protein structures face numerous technical challenges (Benjin and Ling, 2020; McPherson and Gavira, 2014; Montelione et al, 2000); thus, the structural characterization of the *L. pneumophila* effector arsenal remains largely incomplete where more than 40% lack experimental structures and have no functional annotations associated with them in the UniProt database (https://www.uniprot.org). Recently, new neural network (NN) and large language model (LLM)-based computational approaches, such as AlphaFold2 (Jumper et al, 2021), RoseTTAFold (Baek et al, 2021), and Evolutionary-scale modeling (ESM) Fold (Lin et al, 2023), have dramatically improved the accuracy of protein 3D structure prediction from primary sequences. These methods approach the accuracy of experimentally derived molecular structures for single proteins (Bertoline et al, 2023; Janes and Beltrao, 2024; Perrakis and Sixma, 2021), though they still face challenges in predicting multiple conformations, metal, co-factor, or ligand binding, as well as protein–protein interactions. Nevertheless, these methods allow for the analysis of structural models of large, previously uncharacterized protein families (Pinheiro et al, 2021) and entire proteomes (Tunyasuvunakool et al, 2021). Given that a large portion of the *L. pneumophila* effector arsenal remains both structurally and functionally uncharacterized, we conducted a global analysis of structural models generated by AI-based algorithms, particularly AlphaFold2, of all reported *L. pneumophila* effectors. We analyzed them for the presence of globular domains and functionally relevant structural motifs, which we then validated by leveraging the growth defect phenotype caused by the ectopic expression of effectors in the *Saccharomyces cerevisiae* model system. This approach allowed for the dramatic expansion of our functional predictions for the *L. pneumophila* effectorome, including newly predicted cysteine proteases, metalloproteases, kinases, α/β hydrolases, ADP-ribosyltransferases, and glycosyltransferases. We also identified 30 effectors containing new, cryptic domains with no detectable structural similarity to any experimentally characterized 3D structure.

# Results

## *L. pneumophila* effectors carry an extensive repertoire of predicted functional domains

To cast a wide net for this study, we compiled a list of *L. pneumophila* proteins from previously reported genome-wide evaluations of their translocation in a Dot/Icm-dependent manner into the host cell (Burstein et al, 2009; Huang et al, 2011; Zhu et al,

2011). This approach identified 368 *L. pneumophila* effector proteins (Dataset EV1) with 227 of them described by the UniProt database (https://www.uniprot.org) as "uncharacterized" or as "domain of unknown function (DUF)-containing" proteins and another 43 with a name, but no functional annotations.

A search using BlastP (Altschul et al, 1997) against the Protein Data Bank (PDB) repository showed that at the time of this analysis (see Methods), the 3D molecular structures of 41 *L. pneumophila* effectors had been experimentally characterized (Dataset EV1) to their complete or almost complete length (i.e. structure covered more than 90% of primary sequence with more than 90% identical residues). For an additional 44 effectors, molecular structures were available for a portion of the protein (i.e. structures covering less than 90% of primary sequence with more than 90% sequence identity) (Dataset EV2). Of the remaining 283 effectors, 61 shared strong sequence similarity with structurally characterized proteins (Dataset EV1), leaving 222 effectors with no meaningful structural annotations. Consequently, more than half of the overall *L. pneumophila* effector repertoire showed no strong primary sequence similarity with any structurally characterized proteins. We hypothesized that the gap in structural annotations can be now filled by high-quality structural predictions that can be used as a starting point for their functional characterization.

We analyzed Alphafold2 structural prediction models of all *L. pneumophila* effector proteins currently lacking structural coverage along with their primary sequences to identify potential functionally distinct domains (see the Methods section). Each of the predicted domains was classified based on their structural or sequence similarity to known domains in the ECOD (Cheng et al, 2014) and InterPro (Blum et al, 2021) databases, respectively; and, where applicable, previously reported experimental data. Predicted representatives of established functional domain families identified in effector protein models were further analyzed for the presence of known functionally relevant molecular motifs.

Including the AlphaFold2 models dramatically increased the number of predicted domain types identified in *L. pneumophila* effectors, as compared to previous analyses (Burstein et al, 2016; Gomez-Valero et al, 2019; Gomez-Valero et al, 2011) and public databases, such as UniProt (https://www.uniprot.org) (Fig. 1). We identified at least one distinct ECOD-classified domain type or structural motif in the models of 286 *L. pneumophila* effectors, with a maximum of six domains and motifs identified in the case of two effectors (Dataset EV2). For 82 effectors, we could not assign any specific ECOD-classified domain type to the predicted 3D structure. The models of these effectors were either unstructured (disordered), consisted of structural elements, such as α-helical bundles and/or transmembrane helices with no significant structural similarity to a specific ECOD domain, or showed a unique and potentially novel fold. To facilitate the follow-up functional characterization of identified functional domains, we have captured our analysis on a publicly accessible web page: https://pathogens3d.org/legionella-pneumophila.

In line with previous global analyses of *L. pneumophila* effector primary sequences (Burstein et al, 2016; Gomez-Valero et al, 2019), the analysis of effector models confirmed the presence of a significant number of so-called tandem repeat motifs, including armadillo (ARM) identified in 27 models, ankyrin (ANK) identified in 24 models, or leucine-rich repeats (LRR) identified in nine models (Dataset EV2; Fig. 1), all of which are typically associated with protein–protein interactions. Interestingly, ANK and ARM repeats were usually found

in effector models that also contained other domains. In contrast, LRR domains were the only functional element identified in effector models (Dataset EV2). While the presence of most of these structural motifs was also predicted in previous reports using primary sequence-based tools (Burstein et al, 2016; Cazalet et al, 2004), including the 3D models as a starting point significantly expanded the number of effectors predicted to contain these structural elements (Dataset EV2; Fig. 1).

The overall repertoire of distinct domains identified in the analyzed effector models included 157 structurally diverse domain types associated with known enzymatic activities, protein–protein and protein–nucleic acid interaction functions, as well as eukaryotic-specific post-translational modification cascades (Dataset EV2). Effector regions that corresponded to identified structural domains matched the regions modeled with high confidence. In contrast, only 2.4% of residues modeled with a confidence above 70 are outside of identified structural motifs and domains. Of the domain categories present in multiple effectors, the cysteine protease domain represents the most abundant globular domain type—identified in 37 effector models (Fig. 1). The next largest group are protein kinase domains—found in 17 effector models (Fig. 1). Overall, a total of 66 domain types were identified in more than one effector model, while 91 of the predicted domain types were present only in a single *L. pneumophila* effector model (Dataset EV2). We interpret this as an indication of high functional diversity across the *L. pneumophila* effector repertoire. Finally, our analysis identified 35 predicted domains that showed no strong structural similarity to experimentally characterized protein structures, thus, suggesting potential novel structural folds (Dataset EV2). Our analysis also identified a significant number of previously unrecognized domains, including cryptic domains that appear to share no structural similarity with experimentally defined structures. In particular, 199 of the 270 effector proteins annotated in the UniProt database (https://www.uniprot.org) as uncharacterized ("uncharacterized", "domain of unknown function (DUF)-containing" proteins, and proteins with names, but no functional annotations), we have identified a domain that allowed us to provide at least a partial functional annotation. Furthermore, in the case of 15 effector models, we identified cryptic folds in uncharacterized domains, which were the only domains in these proteins, hence we could not assign functions based on structural similarity to a known protein.

Notably, while some models lacking defined secondary structure represented cases of the AlphaFold2 algorithm's failure to generate a high-confidence model, most appear to be confidently predicted to be intrinsically disordered, which included both full-length disordered proteins as well as disordered regions interspaced with structured domains. The prevalence of disordered regions in *L. pneumophila* effector models was notably higher than in the rest of *L. pneumophila* proteome. According to UniProt (https://www.uniprot.org), 24% (90 out of 368) of effectors contain at least one disordered region, while the prevalence of such regions in non-effector proteins is only 4.8%. The higher value for effectors is more typical of eukaryotic proteomes (Basile et al, 2019) and may constitute another "eukaryotic-like" feature of *L. pneumophila* effectors.

## Predicted cysteine protease domains in *L. pneumophila* effectors show functional diversity

Representatives of the cysteine protease domain have been characterized across all kingdoms of life and have been the subject of extensive analysis due to their critical roles in diverse biological

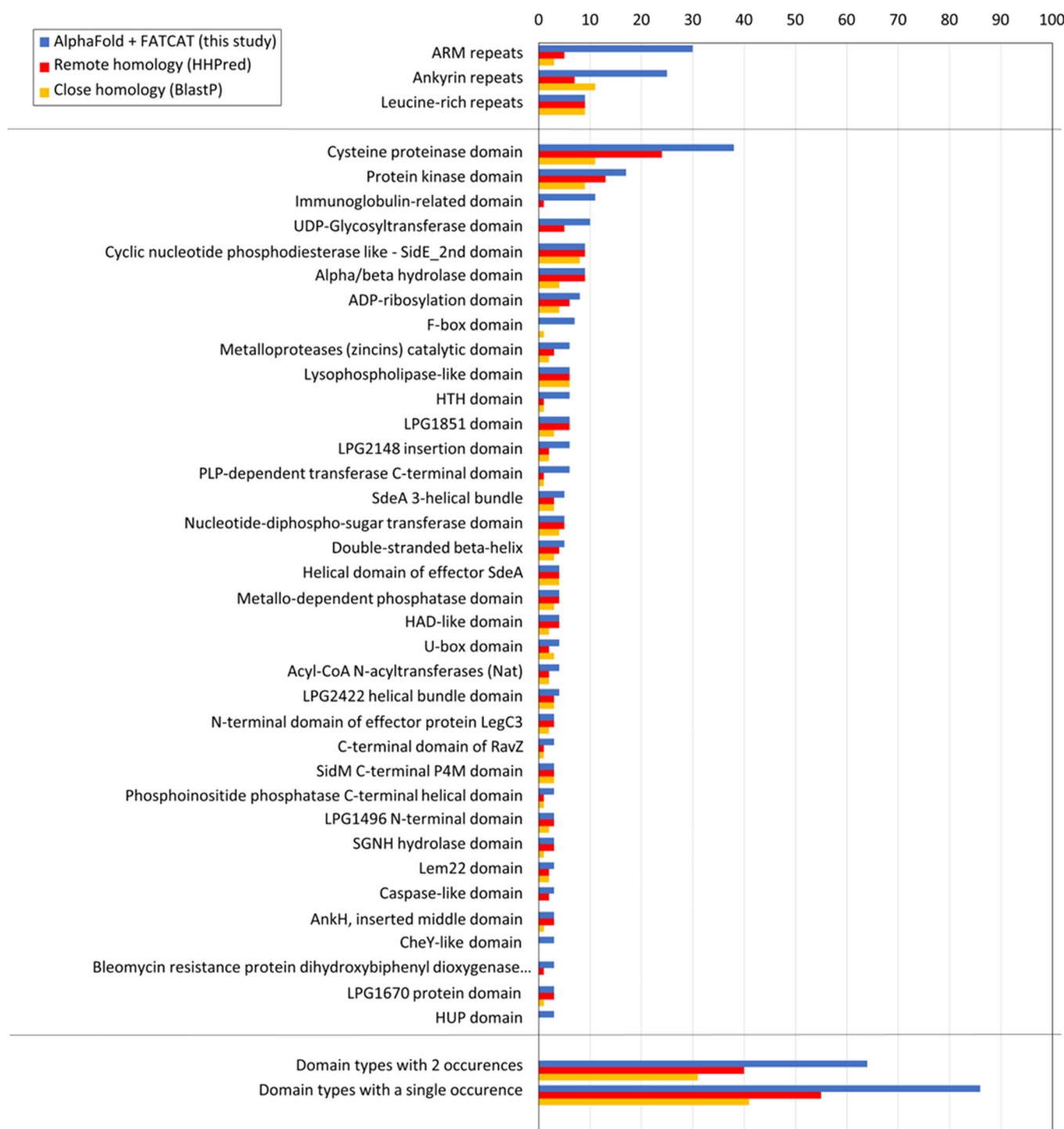

**Figure 1. Functional domain occurrences from 368 *L. pneumophila* effector models.**

The number of occurrences of different ECOD domain types: blue bars—identified using FATCAT in 3D models built with AlphaFold (this study), red bars—identified with remote homology recognition program HHPred, yellow bars—identified by close homology detected with BlastP. As expected, domain-type occurrences identified by homology are a subset of those identified in 3D models.

processes, including protein degradation, cell signaling, and the immune response (Lopez-Otin and Bond, 2008; Verma et al, 2016). These enzymatic domains contain a conserved cysteine residue typically paired with histidine and an aspartate, or asparagine,

residue arranged into a catalytic triad (Rawat et al, 2021). Some cysteine proteases contain a catalytic dyad formed by cysteine and histidine residues (Rawat et al, 2021). The active site containing these catalytic residues is usually located in a cleft

between the two lobes of the α/β domain (Hofer et al, 2020; Verma et al, 2016).

The presence and function of cysteine protease domains have been previously reported for 15 *L. pneumophila* effectors, and the 3D structures of several of these cysteine protease domains have been experimentally defined (Table 1). Expanding on these previous studies, our analysis of effector structural models suggested the presence of cysteine protease domains in 21 additional effectors (Fig. 2A,B; Table 1). In 11 of these, we were able to identify an active site cavity with cysteine, aspartate, and histidine residues arranged in a potential catalytic triad (Table 1). In five effector models, the putative catalytic site featured an asparagine instead of a catalytic aspartate residue (Table 1).

For the Lpg1949/Lem16, Lpg2538, and Lpg2907/MavW models, the predicted cysteine protease domains showed similarity to members of the YopJ effector family (Appendix Fig. S1A). The members of this family - found in human pathogens such as *Yersinia* species, and in several plant pathogens—have only been associated with the type 3 secretion system (T3SS) (Lewis et al, 2011; Ma and Ma, 2016; Meinzer et al, 2012; Orth et al, 2000; Xia et al, 2021). The cysteine protease domain in YopJ effectors demonstrates acetyltransferase activity, which is activated by the eukaryote-specific co-factor inositol hexakisphosphate (IP$_6$) (Mittal et al, 2010). This co-factor binds to a conserved, positively charged pocket on the effector surface (Mittal et al, 2010). Along with the identification of active site pockets consisting of either a Cys-His-Asp/Glu triad or Cys-His dyad (Mukherjee et al, 2006; Orth et al, 2000; Tomar et al, 2023; Zhang et al, 2017), the analysis of the corresponding predicted domains in Lpg1949/Lem16, Lpg2538, and Lpg2907/MavW (Appendix Fig. S1A) suggested the presence of a positively charged pocket typical of IP$_6$ binding (Appendix Fig. S1B). Furthermore, Lpg1949/Lem16 functions as an acetyl-transferase rather than a protease (Hermanns et al, 2020); however, the host substrate and biological significance of this effector during infection remain to be determined. Accordingly, based on our analysis, we hypothesize that along with Lpg1949/Lem16, Lpg2538 and Lpg2907/MavW may also demonstrate YopJ-like acetyltransferase activity; therefore, potentially expanding the YopJ enzyme family to effectors translocated by the *L. pneumophila* T4SS.

Previous studies have identified Lpg0227/LotD, Lpg2248/LotA, Lpg1621/LotB, and Lpg2529/LotC as novel bacterial deubiquitinases, which, due to their primary sequence and structural similarity to eukaryotic ovarian-tumor deubiquitinases (Kang et al, 2023; Kubori et al, 2018; Schubert et al, 2020), have been termed the *Legionella* OTU-like DUBs (Lot-DUBS). Typically, Lot-DUBs harbor a catalytic triad consisting of Cys-His-Asp residues. Based on structural similarity, our analysis suggested that a similar domain may also be present in Lpg2952/Ceg35. However, the Lpg2952/Ceg35 model did not reveal any suitable candidates for the catalytic residues typical of Lot-DUB enzymes (Appendix Fig. S2A,B) and a previous study was not able to demonstrate DUB activity for Lpg2952/Ceg35 (Hermanns et al, 2020), raising the possibility that the activity of Lpg2952/Ceg35 may have deviated from Lot-DUBs.

The cysteine protease domains in the structural models of Lpg1355/SidG, Lpg1387, Lpg1797, and Lpg1909 share structural features with the NlpC/P60 cysteine endopeptidase family, which include known peptidoglycan (PG) degraders (Griffin et al, 2023; Hersch et al, 2020). Notably, the structurally characterized effector TseH from *Vibrio cholerae* translocated by type 6 secretion system

**Table 1. The effector models that have a cysteine protease domain and the coordinates of previously published or predicted catalytic residues.**

| Effector | Potential, or previously described, catalytic residues | Reference |
|---|---|---|
| Lpg0056 | Cys19-His182-Asp200 | This Study |
| Lpg0126/CegC2 | Cys21-His228-Asn226 | This Study |
| Lpg0160/RavD | Cys12-His94-Ser111 | (Wan et al, 2019b) |
| Lpg0196/RavF | Cys320-His32-Ser89 | This Study |
| Lpg0227/Ceg7/LotD | Cys13-His256-Asp6 | (Kang et al, 2023) |
| Lpg0234/SidE | Cys117-His64-Asp80 | (Sheedlo et al, 2015) |
| Lpg0284/Ceg10 | Cys159-His192-Asp204 | This Study |
| Lpg0285/Lem2 | Cys84-His121-Asp133 | This Study |
| Lpg0403/LegA7 | Cys73-His217-Asp55-Asn232 | This Study |
| Lpg0944/RavJ | Cys101-His138-Asp170 | (Urbanus et al, 2016) |
| Lpg1110/Lem5 | Cys109-His137-Asp148 | This Study |
| Lpg1120/Lem6 | Cys23-His205-Asn228 | This Study |
| Lpg1148/LupA | Cys-252-His183-Asp207 | (Urbanus et al, 2016) |
| Lpg1355/SidG | Cys623-His57-Asp158-Glu162 | This Study |
| Lpg1387 | Cys403-His38-Ser104 | This Study |
| Lpg1621/Ceg23/LotB | Cys29-His270-Asp29 | (Ma et al, 2020) |
| Lpg1683/RavZ | Cys258-His176-Asp197 | (Horenkamp et al, 2015) |
| Lpg1797/RvfA | Cys314-His45-Ser102 | This Study |
| Lpg1909 | Cys166-His22 | This Study |
| Lpg1949/Lem16 | Cys174-His105-Asp124 | This Study |
| Lpg1965/LirE/PieC | Cys449-His549-Asp568 | This Study |
| Lpg1966/LirF/PieD | Cys296-His396-Glu416 | This Study |
| Lpg2143 | Cys138-His259-Asp274 | This Study |
| Lpg2147/MavC | Cys74-His231-Gln252 | (Valleau et al, 2018) |
| Lpg2148/MvcA | Cys83-His244-Gln265 | (Valleau et al, 2018) |
| Lpg2153/SdeC | Cys118-His64-Asp80 | (Sheedlo et al, 2015) |
| Lpg2156/SdeB | Cys123-His69-Asp85 | (Sheedlo et al, 2015) |
| Lpg2157/SdeA | Cys118-His64-Asp80-Asn114 | (Sheedlo et al, 2015) |
| Lpg2215/LegA2 | Cys33-His145-Asn162 | This Study |
| Lpg2248/LotA/Lem21 | Domain 1 (residues 1-294): Cys13-His237-Asn239 and Domain 2 (residues 296-544): Cys303-Asn296-His535 | (Takekawa et al, 2022; Warren et al, 2023) |
| Lpg2433/Ceg30 | Cys26-His205-Asp229-Asn226 | This Study |
| Lpg2529/LotC/Lem27 | Cys24-His304-Asp17 | (Shin et al, 2020) |
| Lpg2538 | Cys299-His231 | This Study |
| Lpg2586 | Cys106-His291-Asn320-Gln100 | This Study |
| Lpg2907/MavW | Cys263-His192-Asp215 | This Study |
| Lpg2952/Ceg35 | No functional residues could be assigned | This Study |

(T6SS) (Altindis et al, 2015; Hersch et al, 2020) also belongs to this protein family. While the specific enzymatic activity of TseH remains enigmatic, the TseH structure features two lobes that form a pocket housing glutamate, histidine, and cysteine residues shown to be essential for its activity (Hersch et al, 2020). Similarly, we identified a histidine and a cysteine at similar positions and additional conserved residues in the interlobar pocket of Lpg1355/SidG, Lpg1387, Lpg1797, and Lpg1909 models potentially

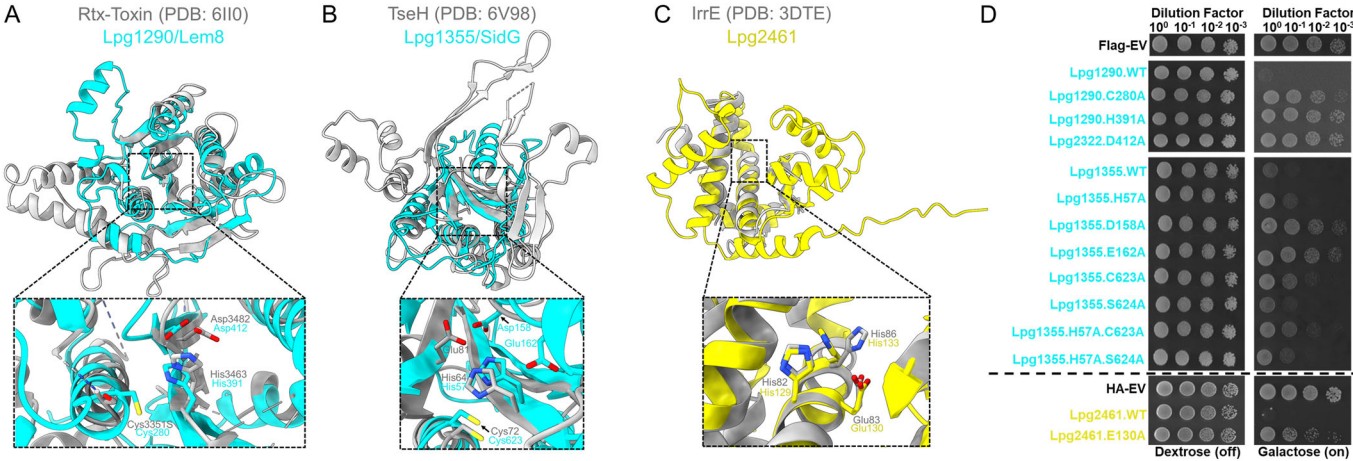

**Figure 2.    Revelation of two previously unrecognized cysteine protease and metalloprotease effectors that are known to cause yeast toxicity.**

(A, B) Structural alignment of the cysteine protease domain of Lpg1290/Lem8 (cyan, residues 244–457) with Rtx-toxin (gray, residues 3309–3562) (Lee et al, 2019), followed by the alignment of the cysteine protease domain model of Lpg1355/SidG (cyan, residues 16–195 + 563–665) with TseH effector (gray, residues 21–223) from *Vibrio cholera* (Hersch et al, 2020). Furthermore, in the case of Lpg1355/SidG, an adjacent α-helix in the predicted catalytic pocket harbors two residues, Asp158 and Glu162, orientated in a position that suggests an involvement in the biochemical activity of this effector. (C) Model of Lpg2461 (yellow, residues 1–212) structurally aligned with IrrE from *Deinococcus derserti* (residues 1–281) (Vujicic-Zagar et al, 2009). Below is a close view of the potential catalytic residues (cyan or yellow sticks) of each effector model determined by their top structural hit (gray sticks) from our FATCAT analysis. (D) Yeast toxicity panel of FLAG-tagged constructs of Lpg1290/Lem8, Lpg1355/SidG, and their respective mutants, followed by the HA-tagged constructs of Lpg2461 and its mutant. Expression of these constructs are found in Appendix Figs. S10, 11. In the case of Lpg1290/Lem8, a Cys280Ala mutation was tested instead of the Cys280Ser mutation used in a previous study. Serial dilutions were spotted on SD media containing either dextrose (repressing) or galactose (inducing). A representative experiment of three independent replicates is shown.

important for corresponding effector activity (Fig. 2B; Appendix Fig. S3). In contrast, we were unable to assign an appropriate third residue in the Lpg1909 model to match the catalytic triad found in TseH or other structurally characterized cysteine proteases, suggesting that this effector potentially relies on a Cys166-His22 dyad for catalysis (Appendix Fig. S3).

Finally, Lpg2586, also identified as a T4SS effector containing a potential cysteine protease domain, shares high sequence similarity (~30%) with another *L. pneumophila* effector, Lpg2622. Lpg2622 has been associated with the *L. pneumophila* type II secretion system (T2SS) and was characterized as a member of the C1 peptidase family (Gong et al, 2018). Like Lpg2622 (PDB: 6A0N), Lpg2586 also contains a unique hairpin-turn-helix motif–shown to be essential for Lpg2622 protease activity and an N-terminal β-sheet–shown to be involved in regulating the activity of Lpg2622 (Appendix Fig. S4A,B) (Gong et al, 2018). Our analysis also suggested that residues Cys106, His291, Asn320, and Gln100 in Lpg2586 may be important for catalysis based on the role of equivalent residues demonstrated for Lpg2622 (Appendix Fig. S4A; Table 1).

## Identification of three additional metalloprotease domain-containing effectors in the *L. pneumophila* arsenal

Metalloproteases are important components of multiple biological processes in eukaryotes. These enzymes facilitate the degradation of extracellular membrane proteins, glycoproteins, growth factors, cytoskeletal proteins, and cytokines—which, in turn, regulate apoptotic, cellular differentiation, and proliferation pathways (de Almeida et al, 2022; Parks et al, 2004; Sternlicht and Werb, 2001). When compared to cysteine proteases, metalloproteases have several distinct structural features essential for their functionality. Typically, these enzymes contain an α-helical pro-domain that

regulates the protease activity and a catalytic domain consisting of roughly five β-strands and three α-helices (Laronha and Caldeira, 2020). The catalytic pocket of metalloproteases is made of an α-helix harboring a highly conserved sequence motif: His-Glu-X-X-His-X, where "X" represents any amino acid (Laronha and Caldeira, 2020; Sternlicht and Werb, 2001). The histidine residues in this motif are required for the coordination of a divalent cation, typically a $Zn^{2+}$, that is involved in peptide bond hydrolysis (Ra and Parks, 2007).

Only two *L. pneumophila* effectors (Lpg0969/RavK and Lpg2999/LegP) have been identified as metalloproteases based on primary sequence analysis (de Felipe et al, 2005; Liu et al, 2017). Lpg0969/RavK was shown to cleave actin to prevent the formation of actin polymers when ectopically expressed in HEK293T cells and during *L. pneumophila* infection. However, the biological significance of this cleavage remains unclear (Liu et al, 2017). While neither the host target of Lpg2999/LegP nor its function as a canonical metalloprotease has been experimentally validated, our analysis confirms the presence of the catalytic motif (Appendix Fig. S5A). Furthermore, our analysis suggested three additional effectors (Lpg0041, Lpg1667, and Lpg2461) harbor a metalloprotease catalytic domain (Fig. 2C; Appendix Fig. S5B; Table 2). The models of these three effectors contain the His-Glu-X-X-His-X motif typical of the metalloprotease fold (Fig. 2C; Appendix Fig. S5A,B; Table 2). Notably, among the five metalloprotease domain-containing effector models, only Lpg0041 and Lpg1667 contained an additional structural element. The Lpg0041 model contains an additional C-terminal structural element composed of the three tandem beta-sandwiches, whereas Lpg1667 is predicted to contain a single β-sandwich spanning residues 57 to 176. We hypothesize that these structural elements contribute to the recognition of the host substrate by these effectors.

**Table 2.** Summary of the predicted and previously published catalytic motifs of the metalloprotease domain-containing effectors identified by the 3D model analysis.

| Effector | Potential, or previously described, catalytic residues | Reference |
|---|---|---|
| Lpg0041 | His350-Glu351-Ile352-Gly353-His354 | This Study |
| Lpg0969/RavK | His95-Glu96-Thr97-Gly98-His99 | (Liu et al, 2017) |
| Lpg1667 | His342-Glu343-Leu344-Gly345-His346 | This Study |
| Lpg2461 | His129-Glu130-Val131-Cys132-His133 | This Study |
| Lpg2999/LegP | His166-Glu167-Ile168-Gly169-His170 | This Study |

## Expansion of the repertoire of *L. pneumophila* effector kinases

Kinases are important mediators of multiple biological functions, such as signal transduction, protein function modulation, and modification of small molecules, including lipids and carbohydrates (Fabbro et al, 2015; Oruganty et al, 2016; Pereira et al, 2011; Rauch et al, 2011). There are multiple molecular folds associated with this biochemical activity, but the most prevalent is the protein kinase fold that adopts a bi-lobed structure comprised of a smaller, all β-strand N-terminal domain and a larger C-terminal mixed α/β domain, with the two domains connected via a short, linear region called the hinge (Arter et al, 2022). The ATP binding site is localized close to the hinge in the cavity formed between the two lobes, with its adenine ring nestled in a hydrophobic pocket and forming hydrogen bonds between its purine nitrogens and residues in the hinge (Arter et al, 2022). In accordance with the catalytic mechanism, the substrate binding site is also found in this cavity, placing it in proximity to the ATP γ-phosphate (Arter et al, 2022). Other salient features of the Protein Kinase fold include the "catalytic loop", which harbors aspartate residues that coordinate $Mg^{2+}$ ions interacting with ATP phosphate oxygens; a glycine-rich loop (also called the "P-loop") that interacts with the ATP phosphate oxygens; and an "activation segment" containing an Asp-Phe-Gly (DFG) sequence and often a tyrosine residue, both of which are important in the regulation of kinase activity as the activation segment is often disordered in the non-phosphorylated state (Arter et al, 2022; Leipe et al, 2003; Nolen et al, 2004; Reinhardt and Leonard, 2023). ATP-grasp is another common molecular fold associated with kinases, which usually act on small molecules (Fawaz et al, 2011). This fold is founded on two α/β domains that bind ATP in the interdomain cleft (Fawaz et al, 2011). We identified 17 effector kinases in *L. pneumophila*, with 15 of these falling into the ECOD T group "Protein Kinase" and two into the ECOD T group "ATP-grasp" (Tables 3 and 4).

Four *L. pneumophila* effector kinases, Lpg0208/LegK4, Lpg1924/LegK7, Lpg2603, and Lpg2975/MavQ, have been structurally characterized (Flayhan et al, 2015; Hsieh et al, 2021; Lee et al, 2020; Sreelatha et al, 2020). Two additional effectors, Lpg1483/LegK1 and Lpg2137/LegK2, although lacking structural characterization, were predicted to harbor a kinase domain through primary sequence analysis and have been shown to exhibit kinase activity (Ge et al, 2009; Hervet et al, 2011; Michard et al, 2015). In addition, *L. pneumophila* has also been shown to repurpose kinase-like folds for novel biochemical activities – as shown in the case of Lpg2155/

SidJ and its paralog Lpg2508/SdjA. Lpg2155/SidJ harbors most of the classical structural and functional features of a kinase, but instead catalyzes the polyglutamylation of the SidE effector family (Adams et al, 2021; Bhogaraju et al, 2019; Black et al, 2019), whereas Lpg2508/SdjA glutamylates and deglutamylates members of the SidE family (Osinski et al, 2021; Song et al, 2021).

Our analysis suggested that 9 additional effectors contain domains falling into the ECOD T group "Protein Kinase" category. Except for Lpg1684, Lpg1924/LegK7, and Lpg1925, all these predicted domains contain the essential structural and catalytic motifs characteristic of kinase enzymatic activity (Table 3; Appendix Fig. S6). Comparative analysis of the effector kinase models with experimentally characterized kinase structures in the PDB suggested that the models of Lpg1483/LegK1 (Ge et al, 2009), Lpg2050 and Lpg2556/LegK3 contain the necessary features of protein kinases (Fig. 3A; Appendix Fig. S6A), while the models of Lpg1316/RavT, Lpg1317/RavW, Lpg2322/AnkK/LegA5 are more structurally similar to kinases targeting lipids (Fig. 3B; Appendix Fig. S6B). Consistent with these predictions, Lpg2322/AnkK/LegA5 has been experimentally shown to be a phosphatidylinositol 3-kinase (PI3K) (Ledvina et al, 2018).

In the remaining effector models of the protein kinase category, we observed significant variation from the canonical motifs established for functional kinases, particularly in the composition of the activation loop and P-loop. Instead of the activation loop DFG motif found in canonical kinases, the models of Lpg1316/RavT, Lpg1408/LicA, and Lpg2322/AnkK/LegA5 featured DHE (residues 206–208), DWE (residues 262–264), and DHD (residues 201–203) sequences, respectively. The P-loop in the Lpg1316/RavT model contained a stretch of three serine residues (residues 77–79), while typical kinases contain at least one glycine at these positions. Lpg1316/RavT shares 45% of sequence identity with Lpg1317/RavW and is predicted to adopt a very similar structure.

The kinase domain identified in the Lpg2050 model is very similar to the structure of the *Shigella* effector OspG (Grishin et al, 2014b; Pruneda et al, 2014) or the *E. coli* effector NleH (Grishin et al, 2014a), both of which require interaction with other proteins to trigger their protein kinase activities (Fig. 3A). The models of Lpg1684 and Lpg1925 show similarity to "ATP-grasp" kinases. In this fold, the ATP binding site is composed of multiple charged residues, including Glu/Asp and Lys/Arg that interact with magnesium ions or the phosphate oxygens of ATP (Fawaz et al, 2011) (Table 4). Both of these effector kinase models contain such charged residues and are most similar in structure to the kinases targeting protein substrates (Appendix Fig. S6C).

Structural elements beyond the "core" kinase domain often contribute to the recognition and positioning of kinase substrates or regulation of kinase activity (Pereira et al, 2011). Specifically, interactions between the effector kinase Lpg1924/LegK7 and host protein Mps one binder kinase activator 1A (MOB1A) involve the N-terminal α-helical domain in addition to the "core" kinase domain (Lee et al, 2020). Therefore, we examined the structural models of effectors with predicted kinase domains for the presence of such additional structural elements. Lpg0208/LegK4, Lpg1483/LegK1, Lpg2322/AnkK/LegA5, Lpg2508/SdjA, and Lpg2556/LegK3 are predicted to harbor their kinase domains in the N-terminal portion of the protein, followed by α-helical bundles; this region in Lpg0208/LegK4 may adopt an ARM fold (Dataset EV2). The Lpg2137/LegK2 model also features the kinase followed by an α-

**Table 3.** A summarization of kinase domain-containing effectors from *L. pneumophila* identified from predicted models and their putative, or previously identified, catalytic residues.

| Effector | Potential, or previously described, catalytic residues | | Reference |
|---|---|---|---|
| | Lys/Asp - Glu pair | DFG motif | |
| Lpg0208/LegK4 | Lys110-Glu125 | Asp219-Phe220-Gly221 | (Flayhan et al, 2015) |
| Lpg1316/RavT | Lys93-Glu103 | Asp203-His204-Glu205 | This Study |
| Lpg1317/RavW | Lys62-Glu73 | Asp169-His170-Glu171 | This Study |
| Lpg1408/LicA | Arg101-Glu116 | Asp259-Trp260-Glu261 | This Study |
| Lpg1483/LegK1 | Lys121-Glu137 | Asp244-Phe245-Gly246 | (Ge et al, 2009) |
| Lpg1924/LegK7 | Arg209-Glu219 | Asp324-Arg325-Lys326 | (Lee et al, 2020) |
| Lpg2050 | Lys57-Glu87 | Asp178-Leu179-Asp180 | This Study |
| Lpg2137/LegK2 | Lys112-Glu128 | Asp223-Ala224-Gly225 | (Hervet et al, 2011; Michard et al, 2015) |
| Lpg2155/SidJ | Lys367-Glu381 | Asp542-Leu543-Gly544 | (Adams et al, 2021; Osinski et al, 2021) |
| Lpg2322/AnkK/LegA5 | Lys43-Glu56 | Asp201-His202-Asp203 | This Study |
| Lpg2508/SdjA | Lys305-Glu319 | Asp480-Leu481-Gly482 | This Study |
| Lpg2556/LegK3 | Lys87-Glu110 | Asp207-Tyr208-Gly209 | This Study |
| Lpg2603/Lem28/SdmB | Lys114-(missing) | Asp225-Leu226-Asp227 | (Sreelatha et al, 2020) |
| Lpg2975/MavQ | Lys46-(missing) | Asp160-Phe161-Asp162 | (Hsieh et al, 2021) |

**Table 4.** The identification of two ATP-grasp kinase domain-containing effectors from *L. pneumophila* and their predicted catalytic residues from this study.

| Effector | Potential, or previously described, catalytic residues | Reference |
|---|---|---|
| Lpg1684 | Arg145-Arg234-Glu244-Asp331-Glu363 | This Study |
| Lpg1925 | Lys229-Lys273-Asn388-Asp401 | This Study |

helical bundle, plus a small α/β domain preceding the kinase domain at the N-terminus. This N-terminal domain is predicted to pack against the β-lobe of the kinase. The model of ATP-grasp effector kinase Lpg1925 also reveals a multi-domain composition, with an α/β structure at its N-terminus, the kinase domain, including an α-helical insert followed by a long α-helical hairpin that packs against the kinase domain, another α/β domain, and finally an α-helical bundle.

## α/β hydrolase domains are recurring in *L. pneumophila* effectors

The specific activities of α/β hydrolases can vary widely between members of this superfamily (Kourist et al, 2010; Nardini and Dijkstra, 1999). The general structure of α/β hydrolases comprises a central six-stranded central β-sheet surrounded by α-helices, with the ligand binding site found at the "top" of the β-sheet where the C-terminal ends of each β-strand align. Members of this family typically harbor a catalytic triad formed of a serine residue fulfilling the role of the catalytic nucleophile, a histidine and an acidic residue (almost always an aspartate), localized to loops between β-strands. An important distinguishing feature of some α/β hydrolases is the presence of a "lid" subdomain inserted into the central β-sheet, which varies in size and structural features (Kourist et al, 2010; Nardini and Dijkstra, 1999). This subdomain often forms part of the ligand binding cleft and can create solvent-excluded pockets for binding hydrophobic compounds

such as lipids. A subgroup, called SGNH hydrolases, share a common three-layer α/β/α structure and unites a group of enzymes with diverse specific activities, including carbohydrate esterase, thioesterase, protease, arylesterase, and lysophospholipase (Anderson et al, 2022).

Our analysis indicated that nine *L. pneumophila* effector models contained the α/β hydrolase domain (ECOD T group "alpha/beta-hydrolases"), and three effector models contained domains reminiscent of SGNH hydrolases (ECOD T group "SGNH hydrolase") (Table 5). Only three of these effectors - Lpg1642/SidB, Lpg1907, and Lpg2911 - have been previously reported to possess an α/β hydrolase domain (Gomez-Valero et al, 2011; Gomez-Valero et al, 2014; Luo and Isberg, 2004). In addition, the structure of Lpg2422/Lem25 has been experimentally determined (PDB 4M0M), confirming the presence of this domain. However, to the best of our knowledge, none of these effector proteins have been experimentally characterized to possess hydrolase activity. Along the same lines, we did not find any previous reports describing *Legionella* effectors possessing the SGNH hydrolase domain.

The α/β hydrolase domains of six effectors (Lpg1108/RavL, Lpg1642/SidB, Lpg1907, Lpg2391/SdbC, Lpg2422/Lem25, Lpg2911) contain potential serine-aspartate-histidine catalytic triads, while the domains predicted in Lpg0275/SdbA, Lpg1959, and Lpg2482/SdbB feature potential cysteine-aspartate-histidine triads (Fig. 3C–E; Appendix Fig. S7A). Our analysis indicates a diversity of lid subdomain structures among these putative α/β hydrolases, while the models of Lpg1108/RavL, Lpg2391/SdbC, and Lpg2422/Lem25 lack the lid subdomain. Lpg2911 shares significant similarity (33% identity) with human cathepsin A (PDB 4AZ0) (Ruf et al, 2012), which suggests that it may possess carboxypeptidase activity. The remaining 8 putative α/β hydrolases show very low sequence identity with structurally and functionally characterized proteins. Some of the predicted features that may provide a cue to the function of these effectors include a long, deep, hydrophobic cleft in the model of Lpg1642/SidB that is reminiscent of the ligand binding site in the monoacylglycerol lipase from *Palaeococcus ferrophilus*

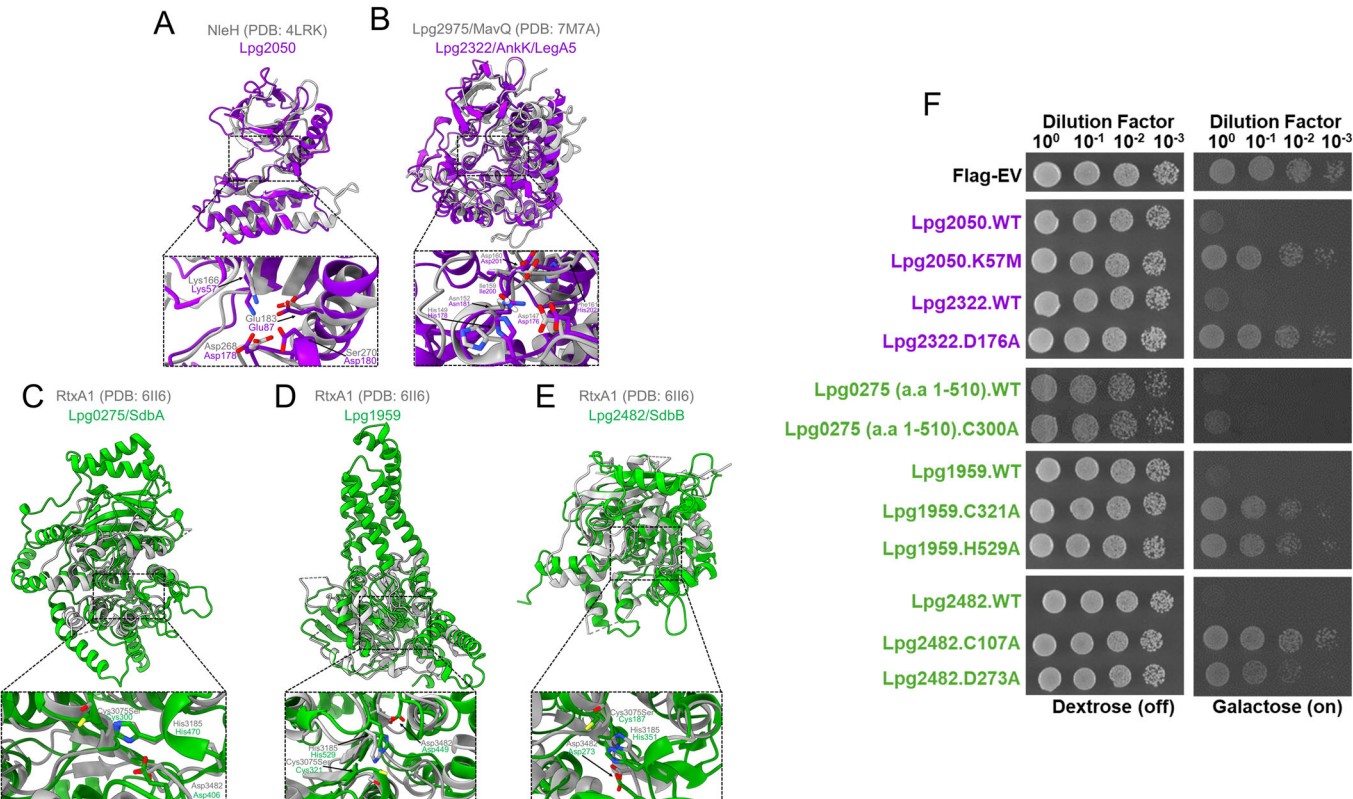

**Figure 3.  Mutagenesis analysis of predicted kinase and α/β hydrolase domains in *L. pneumophila* effector repertoire.**

(A, B) Model of Lpg2050 (purple, residues 1–205) aligned with the structure of the T3SS effector NleH (gray, residues 128-293) (Grishin et al, 2014a). Structural overlay of Lpg2322/AnkK/LegA5's (purple) predicted kinase domain (residues 1–314) onto the lipid kinase domain of Lpg2975/MavQ (gray, residues 1–375) (Hsieh et al, 2021). Putative catalytic residues are shown in the zoomed view. (C–E) Alignments of the effectors possessing α/β hydrolase domains (green) to the top structural hit (gray) from the FATCAT server, followed by the inset view of the putative catalytic residues. (F) The corresponding yeast panel pinpointing potential residues important for the function of these domains in causing yeast toxicity. Effector proteins and their mutants were expressed with an N-terminal FLAG tag (Appendix Fig. S10).

(PDB 6QE2) (Labar et al, 2021), and the model of Lpg2391/SdbC also possesses a similarly broad, wide, and deep cleft.

Lpg0788 and Lpg2587 contained the serine-glycine-asparagine-histidine tetrad typical for the catalytic site of the SGNH-type hydrolase enzymes (Appendix Fig S7B; Table 5). Notably, while the model of Lpg1354 adopts the α/β hydrolase fold, the histidine in the catalytic site is replaced by an aspartate residue, which brings into question whether this enzyme functions as a hydrolase. SGNH hydrolases are further subcategorized into GDSL esterases based on the presence of the corresponding sequence motifs in the N-terminal portion of the protein (Anderson et al, 2022). Accordingly, we identified such motifs for all three of these effectors: residues 71-74 (GSDI) in Lpg0788; residues 8 to 12 (GDSTL) in Lpg1354; and 36 to 39 (GDSY) in Lpg2587. However, reliable prediction of the specific activity for effectors in this group is obscured by low similarity to characterized proteins.

## Predicted ADP-ribosyltransferase domains extend beyond LarT1, Ceg3, and the SidE family in the *L. pneumophila* effector arsenal

Adenosine diphosphate-ribosyltransferases, commonly referred to as ADP-ribosyltransferases, are a group of enzymes with crucial roles in various eukaryotic cellular processes (Luscher et al, 2018). These enzymes catalyze the transfer of ADP-ribose moieties from nicotinamide adenine dinucleotide ($NAD^+$) onto target proteins (Mikolcevic et al, 2021; Suskiewicz et al, 2023). This post-translational modification can modulate protein activity, localization, and interactions within the cell (Suskiewicz et al, 2023). Similar domains have also been characterized as part of bacterial toxins and pathogenic factors, which harness this activity to disrupt critical host cell processes (Dean, 2011; Simon et al, 2014).

In *L. pneumophila*, ADP-ribosyltransferase domains have been characterized primarily as part of the paralogous SidE effector family (Lpg0234/SidE, Lpg2153/SdeC, Lpg2156/SdeB, and Lpg2157/SdeA). These effectors utilize their ADP-ribosyltransferase to ADP-ribosylate (ADPR) the Arg42 residue of ubiquitin, which is then conjugated onto the host substrate by the effector's phosphodiesterase (PDE) domain, forming a host protein-phosphoribosylate-ubiquitin complex (Bhogaraju et al, 2016; Dong et al, 2018; Kalayil et al, 2018; Kotewicz et al, 2017; Qiu et al, 2016), as part of a two-step unorthodox ubiquitination mechanism. In addition to the SidE effector family, ADP-ribosyltransferase domains were also identified in Lpg0080/Ceg3 and Lpg0181. Lpg0080/Ceg3 is an ADP-ribosyltransferase involved in the modification of the Arg236 of the human adenine nucleotide

translocase 2 (ANT2) - which is a membrane-spanning protein required for the exchange of ADP and ATP across the mitochondria inner membrane (Kubori et al, 2022). The modification of ANT2 has also been shown to be reversed by the metaeffector Lpg0081 (Kubori et al, 2022). Lpg0181 targets a conserved Arg residue located in the NAD$^+$ binding pocket of the 120 kDa glutamate dehydrogenase enzyme family present in both fungi and the protist hosts of *Legionella* (Black et al, 2021).

Based on our analysis, the models of two additional effectors - Lpg0796 and Lpg2523/Lem26 - also contain domains similar to an ADP-ribosyltransferase fold (Table 6). A previous study suggested that Lpg2523/Lem26 contains a C-terminal PDE domain based on primary sequence similarity to SdeA (Wan et al, 2019a). However, this study failed to demonstrate Lpg2523/Lem26's ability to hydrolyze ADPR-Ubiquitin, suggesting that this PDE domain may have different substrate specificity (Wan et al, 2019a). Furthermore, the N-terminal domain (residues 5–328) of Lpg2523/Lem26 has limited primary sequence identity to

Lpg0080/Ceg3 (Kubori et al, 2022). Our analysis of the Lpg2523/Lem26 model revealed that apart from structural similarity to the PDE domain of SdeA, the N-terminal domain is reminiscent of the T3SS effector ExoT from *Pseudomonas aeruginosa* (Karlberg et al, 2018) and protein-arginine ADP-ribosyltransferase Tre1 (Ting et al, 2018) that was characterized as part of the T6SS arsenal in the insect pathogen, *Serratia proteamaculans* (Fig. 4A). In line with this analysis, the N-terminal domain of the Lpg2523/Lem26 model features several functionally relevant elements identified in these bacterial effectors, including a catalytic triad consisting of an Arg222-Ser257-Glu294 residues, an ADP-ribosylating turn-turn (ARTT) loop harboring the Glu292-X-Glu294 motif essential for catalysis, followed by a β-sheet involved in the binding and stabilization of NAD$^+$ (Fig. 4A). Overall, the structural analysis of the N-terminal domain of Lpg2523/Lem26 corroborates with the ADP-ribosyltransferase fold prediction.

The analysis of the predicted ADP-ribosyltransferase domain in the Lpg0796 model suggested a structural resemblance to Tse6—a T6SS effector from *P. aeruginosa* (Appendix Fig. S8). Tse6 is an effector that is structurally similar to the catalytic domain of ADP-ribosyltransferase toxins released by human bacterial pathogens—such as the diphtheria toxin from *Corynebacterium diphtheriae* and Exotoxin A from *P. aeruginosa* (Whitney et al, 2015). However, despite having this enzymatic domain, Tse6 has diverged in function by acting as a NAD(P)+ glycohydrolase (Whitney et al, 2015). Our structural analysis of the Lpg0796 model suggests that it possesses a similar conserved β-sheet core involved in NAD+ binding. However, we were not able to identify corresponding residues in the β-sheet core that could contribute to the interaction of this co-factor. Furthermore, the ability of Tse6 to hydrolyze NAD(P)$^+$ is facilitated by Asp396 in an activation loop - a structural element that is absent in Lpg0796 (Whitney et al, 2015) (Appendix Fig. S8). Taken together, we suggest that Lpg0796 lacks catalytically important residues typical of ADP-ribosyltransferases and NAD(P)$^+$ glycohydrolases, suggesting a possible diversification of its biochemical activity.

## Five additional *L. pneumophila* effectors contain potential glycosyltransferase domains

Glycosyltransferases (GT) facilitate the transfer of a glycosidic sugar moiety from a donor co-substrate to an acceptor co-substrate, such as nucleic acids, lipids, and proteins (Zhang et al, 2020). Based on the features of their primary sequence, glycosyltransferases form

**Table 5.** α/β-Hydrolase domain-containing effectors from *L. pneumophila* and their potential catalytic residues that were identified in this study.

| Effector | Potential, or previously described, catalytic residues | GDSL motif | Reference |
|---|---|---|---|
| Lpg0275/SdbA | His470-Cys300-Asp406 | N/A | This Study |
| Lpg1108/RavL | His235-Ser125-Asp206 | N/A | This Study |
| Lpg1642/SidB | His378-Ser190-Asp302 | N/A | This Study |
| Lpg1907 | His378-Ser239-Asp296 | N/A | This Study |
| Lpg1959 | His529-Cys321-Asp449 | N/A | This Study |
| Lpg2391/SdbC | His339-Ser188-Asp270 | N/A | This Study |
| Lpg2422/Lem25 | His235-Ser144-Glu203 | N/A | This Study |
| Lpg2482/SdbB | His351-Cys187-Asp273 | N/A | This Study |
| Lpg2911 | His399-Ser165-Asp341 | N/A | This Study |
| Lpg0788 | His343-Ser73-Asp340 | Gly71-Asp72-Ser73-Leu74 | This Study |
| Lpg1354 | Glu242-Ser10-Ala239 | Gly8-Asp9-Ser10-Leu11 | This Study |
| Lpg2587 | His344-Ser17-Asp341 | Gly15-Asp16-Ser17-Leu18 | This Study |

**Table 6.** The predicted and previously identified catalytic residues of *L. pneumophila* ADP-ribosyltransferase domain-containing effectors from this study and previous ones.

| Effector | Potential, or previously described, catalytic residues | Reference |
|---|---|---|
| Lpg0234/SidE | Arg766-Ser820-Glu860-Ser861-Glu862 | (Qiu et al, 2016) |
| Lpg0080/Ceg3 | Arg44-Ser94-Glu141-Lys142-Glu143 | (Kubori et al, 2022) |
| Lpg0181 | Arg37-Ser86-Glu135-Lys136-Glu137 | (Black et al, 2021) |
| Lpg0796 | No functional residues could be assigned | This Study |
| Lpg2153/SdeC | Arg763-Ser817-Glu857-Asp858-Glu858 | (Kotewicz et al, 2017; Qiu et al, 2016) |
| Lpg2156/SdeB | Arg763-Ser817-Glu857-Asp858-Glu858 | (Qiu et al, 2016) |
| Lpg2157/SdeA | Arg766-Ser820-Glu860-Ser861-Glu862 | (Bhogaraju et al, 2016; Qiu et al, 2016) |
| Lpg2523/Lem26 | Arg222-Ser257-Glu292-Arg293-Glu294 | This Study |

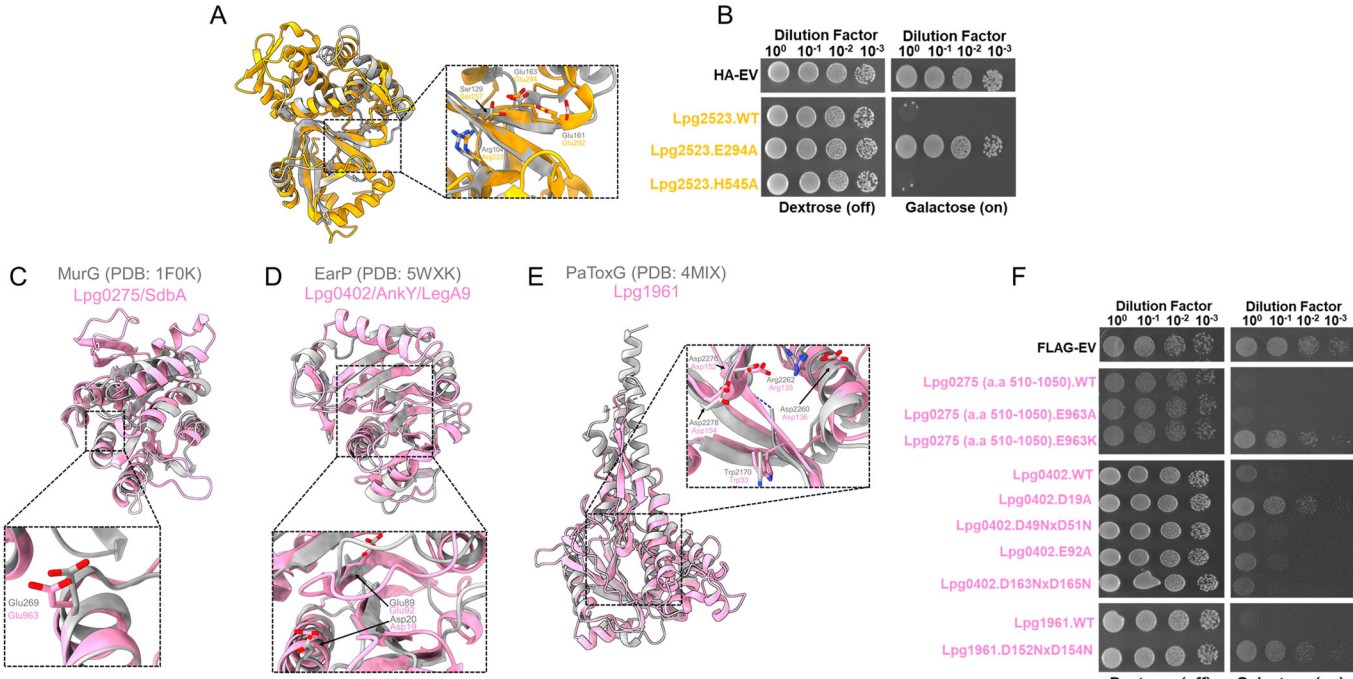

**Figure 4.  The predicted ADP-ribosyltransferase domain of Lpg2523/Lem26 and glycosyltransferase domains of Lpg0275/SdbA, Lpg0402/LegA9, and Lpg1961 are linked to yeast toxicity.**

(A, B) Model of Lpg2523/Lem26's ADP-ribosyltransferase domain (residues 1–324) (gold) overlayed onto the structurally characterized ADP-ribosyltransferase effector Tre1 from *S. proteamaculans* (residues 8–192, gray) (Ting et al, 2018). Residues potentially involved in the catalysis of this predicted ADP-ribosyltransferase domain are also shown in the zoomed-in panel. In the following panel, Glu294 of the ExE motif and His545 of the predicted phosphodiesterase domain were tested for their roles in causing yeast toxicity. The expression of Lpg2523/Lem26 and its mutants are found in Appendix Fig. S11. (C–F) Representation of the glycosyltransferase domain models (pink) from Lpg0275/SdbA (residues 511–1050), Lpg0402 (residues 1–399), and Lpg1961 (residues 27–328) aligned with their top structural hit from the FATCAT analysis. Predicted residues involved in catalysis were assessed in the yeast toxicity panel. The expression level of each construct is found in Appendix Fig. S10.

more than a hundred distinct families and adopt one of the three different structural folds, known as GT-A, GT-B, and GT-C (Rosen et al, 2004). Enzymes classified as GT-A possess a Rossman-like fold with a conserved aspartate-X-aspartate motif required for the coordination of divalent cations and transferase activity (Persson et al, 2001; Taujale et al, 2020). In the case of the GT-B domain, two Rossman-like folds form a central cleft, serving as the site for catalysis (Breton et al, 2006). Furthermore, GT-B enzymes do not require divalent cations for catalysis, thus, lacking the aspartate-X-aspartate motif (Both et al, 2011; Li et al, 2007). Instead, a single negatively-charged (aspartate or glutamate) residue, typically found in the N-terminal Rossman-like fold, has been demonstrated to be essential for catalysis (Breton et al, 2006; Rini et al, 2009). The C-terminal Rossman-like fold is involved in the recognition and binding to the donor co-substrate (Breton et al, 2006). GT-C fold-containing enzymes have a single Roseman-like fold connected to multiple transmembrane helices and use lipid-linked sugars as the donor co-substrate (Alexander and Locher, 2023; Rini et al, 2009).

Several *L. pneumophila* effectors have been demonstrated to be GT-A and GT-B glycosyltransferases, including the members of the Lgt effector family (Lpg1368/Lgt1, Lpg2862/Lgt2, and Lpg1488/Lgt3). The GT-A domain of Lpg1368/Lgt1 is involved in the glycosylation of the human elongation initiation factor A1, which causes protein translation to be inhibited (Belyi et al, 2009; Belyi et al, 2008; Lu et al, 2010). Lpg1978/SetA also contains a GT-A

domain that was shown to have activity against the human transcription factor EB, histones H3.1 and H4 (Beck et al, 2020; Jank et al, 2012). Another GT-A domain-containing effector is Lpp0356/LtpM, which has an atypical active site architecture, consisting of an aspartate-X-asparagine motif required for catalysis (Levanova et al, 2019). The substrate specificity of this effector remains unknown; however, it has been postulated that LptM is involved in hijacking the microtubule vesicle trafficking pathway (Levanova et al, 2019). Lpg2504/SidI is an effector with a GT-B fold that acts as a mannosyltransferase on host ribosomes, resulting in the inhibition of protein translation and the activation of host stress response kinases that promote the transcription of genes involved in cell death (Joseph et al, 2020; Subramanian et al, 2023).

We identified potential glycosyltransferase domains in five additional effectors: Lpg0275/SdbA, Lpg0402/LegA9, Lpg0770, Lpg1151, and Lpg1961 (Fig. 4C–E; Appendix Fig. S9). Lpg1961 has a structural resemblance to the GT-A domain containing PaToxG toxin from the insect and human bacterial pathogen, *Photorhabdus asymbiotica* (Costa et al, 2009; Jank et al, 2013). The model includes an active site strikingly similar to PaToxG (Jank et al, 2013) and contains a typical aspartate-X-aspartate motif, including several residues possibly involved in the interaction and transfer of the donor co-substrate (Fig. 4E). Lpg0275/SdbA, Lpg0402/LegA9, Lpg0770, and Lpg1151 are likely members of the GT-B family with two Rossman-like folds indicative of

**Table 7. List of effectors with a glycosyltransferase domain and their catalytic residues determined from our 3D modeling analysis in this study or functionally characterized in previous studies.**

| Effector | Potential, or previously described, catalytic residues | Reference |
|---|---|---|
| Lpg0275/SdbA | Glu963 | This Study |
| Lpp0365/LptM | Asp140-Thr141-Asn142 | (Levanova et al, 2019) |
| Lpg0402/LegA9 | Asp19 or Glu92 | This Study |
| Lpg0770 | Thr118, Thr170, Glu171, or Asp172 | This Study |
| Lpg1151 | Glu142 or Asp144 | This Study |
| Lpg1386/Lgt1 | Asp246-Ile247-Asp248 | (Belyi et al, 2006) |
| Lpg1488/Lgt3 | Asp292-Ile293-Asp294 | (Belyi et al, 2008) |
| Lpg1961 | Asp152-Thr153-Asp154 | This Study |
| Lpg1978/SetA | Asp134-Ser135-Asp136 | (Beck et al, 2020) |
| Lpg2504/SidI | Arg453-Glu482-Lys600-D724 | (Joseph et al, 2020; Machtens et al, 2023; Subramanian et al, 2023) |
| Lpg2862/Lgt2 | Asp398-Ala399-Asp340 | (Belyi et al, 2008) |

The effectors are either from *L. pneumophila* (Lpg) or *L. pneumophila* Paris strain (Lpp).

glycosyltransferase activity (Fig. 4C,D; Appendix Fig. S9). These effectors likely harbor potential catalytic residues in their N-terminal Rossman-like fold and use their C-terminal fold for nucleotide binding, which is consistent with the functionality of most GT-B enzymes (Table 7).

## Predicted functional domains of *L. pneumophila* effectors manifest in the yeast model system

The ectopic expression of individual *L. pneumophila* effector proteins in *Saccharomyces cerevisiae* often leads to growth defects. This phenomenon has been exploited for the elucidation of the biochemical activity of several effectors in *L. pneumophila* (Belyi et al, 2012; Bhogaraju et al, 2016; Campodonico et al, 2005; de Felipe et al, 2008; Fu et al, 2022; Gaspar and Machner, 2014; Guo et al, 2014; Heidtman et al, 2009; Qiu et al, 2016; Shohdy et al, 2005; Urbanus et al, 2016; Viner et al, 2012), but for most effectors, the structural basis of this "yeast toxicity" remains undefined. Identification of potential functional domains in the 3D models of these uncharacterized effectors provided us with the opportunity to experimentally test the role of these domains in the observed yeast growth defect phenotype.

Our analysis identified predicted functional domains in eleven *L. pneumophila* effectors previously demonstrated to inhibit yeast growth. Correspondingly, we probed the role of individual residues within these predicted domains by site-directed mutagenesis. The expression of Lpg1290/Lem8 was used as a control for this experiment (Fig. 2A,D). The toxicity of this effector to yeast was linked to its cysteine protease domain (Fig. 2A) and was shown to be alleviated by Cys280Ser, His391Ala, or Asp412Ala substitutions of catalytic triad residues in a previous study (Song et al, 2022).

The first effector we tested was Lpg0275/SdbA, where the model suggested the presence of two distinct functional domains connected by a flexible linker (Fig. EV1A,B). Residues 1 to 510 of Lpg0275/SdbA are predicted to form an α/β hydrolase domain, while residues 511 to 1050 are predicted to form a domain adopting a fold reminiscent of GT-B glycosyltransferases (Figs. 3C and 4C). To test if each of these predicted domains contributed to toxicity in yeast, we expressed the corresponding fragments of Lpg0275/SdbA individually in yeast, along

with the variants carrying substitutions in putative catalytic residues in each of the predicted domains. Based on our results, the individual expression of each of the two predicted domains in Lpg0275/SdbA causes toxicity in yeast (Figs. 3C,F and 4C,F), suggesting that both domain activities contribute to this phenotype. The Cys300Ala substitution of the α/β hydrolase domain had a minimal reduction in toxicity compared to the wild-type expression (Fig. 3C,F). The lysine substitution of Glu963 also partially alleviated the growth defect caused by the expression of the Lpg0275/SdbA [511–1050] fragment (Fig. 4C,F). Notably, the alanine substitution of this residue was insufficient to alleviate the toxicity of this fragment. In agreement with these results, the Glu963Lys mutation in the context of full-length Lpg0275/SdbA also partially alleviated toxicity, whereas the double substitution of Cys300Ala and Glu963Lys restored the growth of yeast comparable to that of the FLAG-only yeast control (Fig. EV1C).

Next, we targeted potential catalytic residues in the GT domains predicted in Lpg0402 and Lpg1961 (Fig. 4D,E). In Lpg0402, we identified Asp19 and Glu92 as residues that correspond to the catalytically important residues in experimentally characterized members of this protein family (Fig. 4D). Additionally, our analysis also suggested that Lpg0402 Asp49/Asp51 or Asp163/Asp165 pairs can form catalytically important Asp-X-Asp motif typically found in GT-A enzymes. The yeast toxicity assay suggests that the alanine substitution of Asp19 completely alleviates the toxicity of Lpg0402 which agrees with the suggested role of this residue in the predicted GT-B domain of this effector (Fig. 4F). In contrast, Lpg0402 variants carrying Asp49Gln/Asp51Gln or Asp163Gln/Asp165Gln double mutations had a yeast toxicity profile comparable to the wildtype effector (Fig. 4F).

Similarly, our analysis of the Lpg1961 model suggested residues Asp152 and Asp154 as candidates for the Asp-X-Asp motif essential for the catalytic activity of its predicted GT-A domain (Fig. 4E). The double substitution of these residues to asparagine led to complete alleviation of toxicity, thus, corroborating our structural prediction analysis and suggested role of this predicted domain in the observed phenotype (Fig. 4F).

We next tested the role of the α/β hydrolase domains predicted in the Lpg1959 and Lpg2482/SdbB effectors (Fig. 3D,E). In the case of Lpg1959, we predicted that the residues Cys321, Asp346, and

His529 to fulfill the role of a catalytic triad (Fig. 3D). In support of this, alanine substitution of either Cys321 or His529 resulted in the recovery of yeast growth, clearly implicating the activity of this predicted functional domain to toxicity (Fig. 3F). Similarly, alanine substitution of either Cys107 or Asp273 identified as potential catalytic residues in Lpg2482/SdbB also resulted in the alleviation of toxicity (Fig. 3F).

Our analysis suggested the presence of a kinase domain in Lpg2050 and Lpg2322/AnkK/LegA5 (Fig. 3A,B). Our analysis suggested that Lys57 is catalytically important for Lpg2050 (Fig. 3A) and its substitution to methionine alleviated toxicity (Fig. 3F). A previous study used primary sequence analysis to suggest that His178 makes part of the kinase activation loop of Lpg2322/AnkK/LegA5 (Ledvina et al, 2018). Our analysis suggested that Asp176 might also form part of the kinase activation loop (Fig. 3B), a hypothesis supported by our observation that an Asp176Ala substitution led to the abrogation of Lpg2322/AnkK/LegA5 toxicity in yeast (Fig. 3F).

The model of Lpg1355/SidG suggested the presence of a potential cysteine protease domain, with residues Cys623, Asp158, and His57 forming a canonical catalytic triad (Fig. 2B). The alanine substitution of Asp158 led to the complete alleviation of toxicity, whereas similar substitutions of His57 or Cys623 led to a partial alleviation of the yeast growth defect (Fig. 2D). Our analysis of the Lpg1355/SidG model suggested that the residues Glu162 and Ser624 might also be important for catalytic activity (Fig. 2B). While the alanine substitution of Glu162 led to complete alleviation of yeast toxicity, a similar substitution to Ser624 failed to restore yeast growth (Fig. 2D). Testing the effect of a double His57Ala/Ser624Ala substitution on the toxicity of Lpg1355/SidG, we observed partial restoration of growth comparable to the effect observed for the His57Ala variant. The His57Ala/Cys623Ala double substitution, resulted in a restoration of yeast growth that appeared stronger than either His57Ala or Cys623Ala substitutions alone (Fig. 2D).

We identified a predicted domain reminiscent of zinc metalloproteases in the model of Lpg2461 that included a canonical catalytic motif. An alanine substitution in one of the residues of this motif (Glu130) resulted in the complete restoration of yeast growth (Fig. 2C,D).

The model of Lpg2523/Lem26 revealed two distinct domains connected by a central helical bundle. The N-terminal domain spanning residues 1 to 337 resembles an ADP-ribosyltransferase domain (Fig. 4A), while the model of the C-terminal portion of this effector that spans residues 494 to 779 shares structural similarity to the PDE domain of the SidE effector family. To test if either domain contributes to the toxic effect of Lpg2523/Lem26 in yeast, we targeted Glu294 residue, which is suggested to be part of the Glu-X-Glu motif in the ARTT loop of the ADP-ribosyltransferase domain, and His545 as a putative catalytic residue in the PDE domain. The Glu294Ala (ARTT) substitution abrogated toxicity, whereas the His54 Ala (PDE) substitution did not (Fig. 4B). These results suggested that while the function of the PDE domain of Lpg2523/Lem26 remains enigmatic (Wan et al, 2019a), the predicted ADP-ribosyltransferase domain is responsible for the toxicity phenotype in yeast (Fig. 4B).

## L. pneumophila effector models contain a significant number of cryptic domains, some of which are responsible for toxicity in yeast

A number of predicted structural domains in *L. pneumophila* effector models demonstrate no significant structural similarity to the experimentally defined protein structures deposited to either the ECOD domain or the PDB databases, when analyzed with standard structure comparison tools, such as Dali (Holm, 2022) and FATCAT (Ye and Godzik, 2003). Overall, we identified 35 such "cryptic" domains in 30 effectors, with models of five effectors—Lpg1426/VpdC, Lpg1978/SetA, Lpg1925/CegL1, Lpg1963/PieA, and Lpg1964/PieB— containing two cryptic domains each (Table 8). Notably, most of the effectors from this category are conserved across *Legionella* species, with six effectors also carrying strong similarity to proteins present in intracellular bacteria from the *Coxiellacae* family, such as *Aquicella siphonis*, *Coxiella burnetii*, and *Rickettsiella* spp (Table 8).

The structures of three effectors—Lpg1083/SidN, Lpg1978/SetA, and Lpg2504/SidI—with predicted cryptic domains were experimentally determined and deposited in the PDB database (PDB: 7YJI, PDB: 7TOD, and PDB: 8BVP, respectively) during the course of our analysis (Beck et al, 2022; Gao et al, 2023; Subramanian et al, 2023), confirming the structural predictions and unique fold assignments. Interestingly, manual analysis of these effector structures identified potential distant structural similarities, providing the first indication of their molecular function (Table 8). In the case of Lpg1978/SetA, the cryptic fold is present in the C-terminal domain and consists of an α-helical bundle connected to a β-sheet that forms a positively charged pocket important for interactions with phosphoinositol-3-phosphate (Beck et al, 2022). This domain with a cryptic fold is essential for the localization of Lpg1978/SetA on the surface of the LCV (Beck et al, 2022). The Lpg1083/SidN structure was revealed to be a cryptic domain where the N-terminal region, described to be "paw-like", aids in the localization of the effector to the nucleus (Gao et al, 2023). Furthermore, Lpg1083/SidN is shown to disrupt the lamina complex, which leads to the destabilization of the nuclear envelope (Gao et al, 2023). Awaiting structural characterization of the remaining effector proteins with cryptic predicted structural elements, we used the yeast toxicity model to investigate their functional relevance in Lpg1154/RavQ, Lpg1426/VpdC, Lpg1489/RavX, or Lpg2527/LnaB (Li et al, 2022; Urbanus et al, 2016). Detailed analysis of the structural models combined with primary sequence conservation suggested residues that may be part of the activity of these proteins (Figs. 5A–D and EV2–5). Accordingly, we probed their relevance for toxicity using site-directed mutagenesis (Fig. 5E).

Based on the structural model of Lpg1154/RavQ, residues 59 to 349 form a cryptic α/β domain, with the preceding N-terminal portion to be disordered (Fig. EV2A). This cryptic domain is predicted to contain a three-stranded β-sheet with a two-stranded β-sheet packed onto one face. The other face of the three-stranded β-sheet is packed against a four-helix bundle, and the model also contains five other α-helices. The overall shape of the model is a "T" shape, with the base of the shape formed by an N-terminal α-helix in one direction and the C-terminal three α-helices in the other direction (Fig. EV2A). One of the vertices of the "T" shape is lined up with negatively and positively charged residues at its base and on its side, respectively (Fig. EV2A). According to the comparative sequence analysis using the ConSurf server (Ben Chorin et al, 2020), the residues forming this groove show complete conservation across orthologs found in other *Legionella* species (Fig. EV2B). Specifically, a highly conserved histidine residue (His169 in Lpg1154/RavQ) is positioned at the center of the groove surrounded by other conserved residues, including Asn141, Gln151, Asp218, Glu221, and Arg225 (Fig. EV2B). Accordingly,

Table 8. Residue boundaries of the cryptic domain in *L. pneumophila* effectors and the primary sequence similarity to the closest structural homolog outside of the *Legionella* genus.

| Effector | Amino acid boundary of the cryptic domain (s) | Domains classified in ECOD (for details see Dataset EV1 and Dataset EV2) | Closest homolog to the cryptic domain | Sequence similarity to homolog (%) |
|---|---|---|---|---|
| Lpg0012/CegC1 | 1–474 | - | GGR11_001436 from *Brevundimonas mediterranea* | 20.7 |
| Lpg0172 | 1–109 | Hom (152–211) | BHQ10_008315from *Talaromyces amestolkiae* | 24.0 |
| Lpg0519/Ceg17 | 179–285 | - | AQUSIP_07400from *Aquicella siphonis* | 48.8 |
| Lpg0717 | 24–153 | - | AQUSIP_07790from *Aquicella siphonis* | 52.5 |
| Lpg0733/RavH | 159–272 | LPG2148ins (273–364) | N/A | N/A |
| Lpg1083/SidN | 1–227 | - | CULFYP111_01230 from *Campylobacter ureolyticus* | 53.3 |
| Lpg1154/RavQ | 56–360 | - | AQUSIP_02370from *Aquicella siphonis* | 55.5 |
| Lpg1426/VpdC | 1–239 + 271–299 + 623–678 | PhospholLip (300–622) | FDP41_009687 from *Naegleria fowleri* | 16.2 |
| Lpg1489/RavX | 93–288 | - | N/A | N/A |
| Lpg1551/RavY | 143–241 | - | N/A | N/A |
| Lpg1588/LegC6 | 165–479 | - | N/A | N/A |
| Lpg1692 | 283–421 | Fbox (1–63), Ank (70–140 + 268–282), and LPG2148ins (141–267) | orf 309 from *Coxiella burnetii* | 33.0 |
| Lpg1716 | 1–103 | - | ROA7023_03573 from *Roseisalinus antarcticus* | 6.50 |
| Lpg1925 | 110–320[a] and 321–455 | DnaD (1–109) and Ank(529–821) | GRS96_03500 from *Rathayibacter* sp. VKM Ac-2803[a] | 11.7 |
| Lpg1963/LirC/PieA | 187–320 and 522–681 | - | N/A | N/A |
| Lpg1964/LirD/PieB | 60–225 and 226–404 | - | N/A | N/A |
| Lpg1978/SetA | 282–450 and 515–628[a] | NucDiStran (1–223) | AVI55_00990 from *Piscirickettsia salmonis*[a] | 26.1 |
| Lpg2073 | 250–380 | Imm (5–249) and HEPN (381–535) | BGO78_14705 from *Chloroflexi bacterium* 44-23 | 44.6 |
| Lpg2160 | 1–332 | - | IMCC3317_17790 from *Kordia antarctica* | 14.9 |
| Lpg2207 | 263–407 | LDtrans (43–233) | A3F10_03715 from *Coxiella burnetii* | 65.9 |
| Lpg2223 | 1–384 | - | N/A | N/A |
| Lpg2239 | 513–730 | LPG2422hbun (1–153), SidE2 (158–460), Ank (731–1000), and Ank(1001–1285) | N/A | N/A |
| Lpg2372 | 279–422 | CheY (107–278) | N/A | N/A |
| Lpg2385 | 1–156 | PlnPhC (466–595) | N/A | N/A |
| Lpg2410/VpdA | 23–51 + 404–459 | PhospholLip (53–403) and WYR (513–655) | COA94_02160 from *Rickettsiales bacterium* (unclassified) | 22.7 |
| Lpg2452/LegA14/Ceg31/AnkF | 1–202 + 309–450 | EF (208–299) and Ank (485–921) | N/A | N/A |
| Lpg2504/SidI | 52–137 + 228–247 | UDPGluco(1–51 + 427–568) | N/A | N/A |

**Table 8.** (continued)

| Effector | Amino acid boundary of the cryptic domain (s) | Domains classified in ECOD (for details see Dataset EV1 and Dataset EV2) | Closest homolog to the cryptic domain | Sequence similarity to homolog (%) |
|---|---|---|---|---|
| Lpg2527/LnaB | 94–311 | - | A7I50_5968 from *Pseudomonas antarctica* | 26.8 |
| Lpg2638/MavV | 1–280 | - | N/A | N/A |
| Lpg2912 | 1–122 | AcylC (123–343) | N/A | N/A |

aEffector models with two cryptic domains, followed by an indication of the fold with sequence similarity with another microorganism.

we tested all six of these highly conserved residues, identified to be potentially relevant to Lpg1154/RavQ activity, by mutagenesis. Substitution of His169 to alanine or arginine led to the complete alleviation of yeast toxicity (Fig. 5A,E). The substitution of Glu151 and Asn214 to arginine residues partially alleviated the toxicity of Lpg1154/RavQ (Fig. 5E). In contrast, the substitution of Asp218 and Glu221 to arginine did not rescue yeast growth, suggesting that these substitutions are not detrimental to RavQ's activity in yeast (Fig. 5E). Notably, we also identified orthologs of Lpg1154/RavQ in more distant members of the *Legionellales* order; for instance, *Aquicella siphonis* from the *Coxiellaceae* family but also intracellular pathogens from the *Chlamydiales* order, such as *Waddlia chondrophila* or *Estrella lausannensis* (Appendix Table S1).

The model of Lpg1426/VpdC contained three distinct globular domains: two cryptic domains (residues 1-299 and 622-677) that pack onto each other to potentially form a single functional domain, a lysophospholipase domain (E-Cod T group FabD/lysophospholipase-like, Patatin Family) spanning residues 299 to 621, followed by a predicted disordered region (residues 678–719), and a C-terminal helical bundle that corresponds to residues 720 to 853 (Fig. EV3A). A recent report demonstrated that this effector relies on the C-terminal helical bundle to bind ubiquitin, which, in turn, results in a conformational change to activate its phospholipase domain to facilitate the conversion of phospholipids into lysophospholipids (Li et al, 2022). The activity of Lpg1426/VpdC was shown to be important for LCV expansion during the infection of U937 human macrophages (Li et al, 2022). Together, the predicted cryptic domains in Lpg1426/VpdC form a central seven-stranded β-sheet which is bounded by eight α-helices on one face of the β-sheet, and a small region containing a two-stranded β-sheet and one α-helix on the other face (Fig. EV3A). Further analysis of this cryptic domain suggested distant homology to the Ntox11 domain, a structurally uncharacterized putative toxin domain that is broadly distributed in bacterial and some eukaryotic pathogens (Fig. EV3B) (Zhang et al, 2012). Interestingly, the latter group includes *Naegleria fowleri* amoeba species, which has been identified as one of the natural protist hosts of *Legionella* (Fig. EV3B) (Boamah et al, 2017; Fields, 1996; Newsome et al, 1985). Our analysis of this cryptic domain in Lpg1426/VpdC suggests the formation of a negatively-charged groove formed by highly conserved residues - Arg59, Arg66, E106, E144, Arg184, and Tyr232—present in Lpg1426/VpdC orthologs encoded by the other *Legionella* species (Fig. EV3C,D). Substitutions of Arg66 or Arg184 with glutamate partially attenuated the toxic effect of Lpg1426/VpdC in yeast; thus, in line with this cryptic domain's activity contributing to this phenotype (Fig. 5B,E).

A previous report placed Lpg1489/RavX (Barry et al, 2013) within a large group of effectors (Lpg0103/VipF, Leg0208/LegK4, Lpg0437/Ceg14, Lpg1368/Lgt1, Lpg1488/Lgt3, Lpg2504/SidI, and Lpg2862/Lgt2) shown to manipulate eukaryotic protein translation (Belyi et al, 2006; Belyi et al, 2008; Fontana et al, 2011; Joseph et al, 2020; Moss et al, 2019; Subramanian et al, 2023; Syriste et al, 2024). However, the structural and biochemical basis of this process remains undefined. Our analysis of the structural model of Lpg1489/RavX suggested the presence of a cryptic domain encompassing residues 89 to 267. However, the AlphaFold2 model of Lpg1489/RavX had low (<50%) confidence, which can be likely explained by the fact that this effector is found only in *L. pneumophila*). To overcome the potential limitation of our

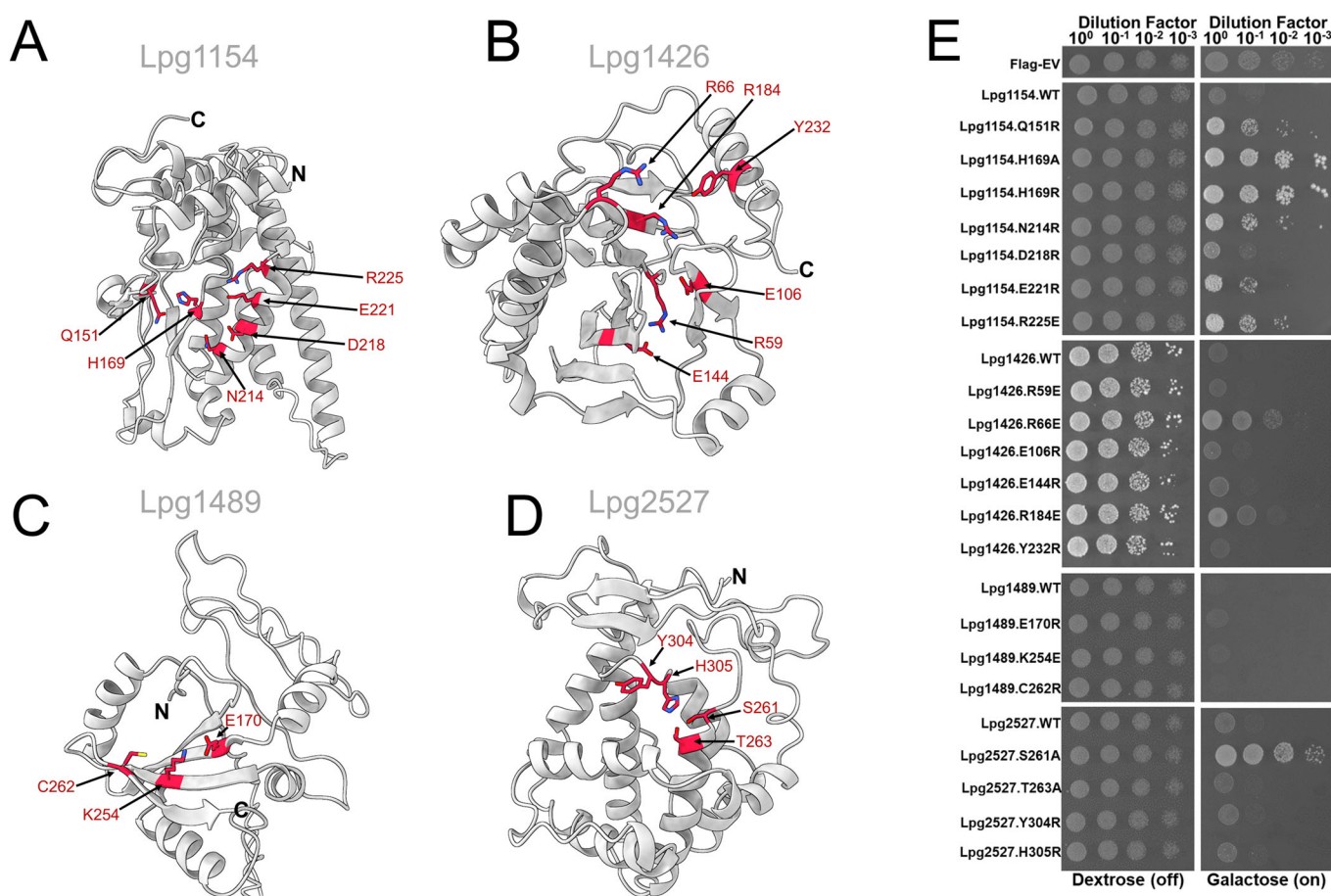

**Figure 5. Cryptic domains in *L. pneumophila* effectors contribute to the yeast toxicity phenotype.**

(A–D) Structural models of the cryptic domains from *L. pneumophila* effectors that are toxic to yeast when expressed ectopically (gray). Residue boundaries of each cryptic domain are described in Table 8. Potentially important residues in the cryptic domains that were mutated and tested in (E) were shown in sticks (red). (E) Yeast spot dilution assay of cryptic domain containing effectors and their respective mutants. The protein expression levels of the mutants tested were analyzed using western blot (Appendix Fig. S12).

modeling approach using Alphafold2, we generated another structural model using ESMFold, an LLM-based algorithm that does not rely on multiple sequence alignment for the prediction (Lin et al, 2023). The ESMFold-generated model of Lpg1489/RavX was similar to the cryptic domain suggested by Alphafold2 but had higher confidence (Fig. EV4A), and was thus selected for further analysis. Interestingly, this model revealed distant structural similarity to *E. coli* enterotoxin (PDB: 1lta) (Merritt et al, 1994), with an RMSD of 3.05 Å over 105 equivalent positions (Fig. EV4B). However, we were unable to assign equivalent residues to the active site of enterotoxin. Additionally, since Lpg1489/RavX is only present in *L. pneumophila*, and has no detectable homologs at the structural and primary sequence levels, we then opted to test three residues—Glu107, Lys254, and Cys262—that resemble a potential catalytic triad in an exposed pocket (Fig. 5E). Despite these predictions, individual substitutions of these residues were unable to alleviate yeast toxicity, suggesting that each is insufficient on its own to alter the activity of RavX (Fig. 5E).

Lpg2527/LnaB is an effector involved in the activation of the NF-κB pathway during infection of HEK293T cells and bone marrow macrophages (Losick et al, 2010). The molecular

mechanism of this activation remains unknown. However, it has been demonstrated that residues 361 to 410 which are suggested to form a potential helical bundle, are important for protein–protein interactions (Losick et al, 2010). The full-length model of Lpg2527/LnaB features a helical bundle (residues 1–110) that leads into a compact cryptic domain spanning residues 111 to 333 (Fig. EV5A). Additional structural elements in the model include a long α-helix (residues 339–393), a small three-helical bundle backed against the long helix (residues 397–497), and an extended disordered region to its C-terminus. The cryptic domain in Lpg2527/LnaB consists of two helical bundles packed against each other and interrupted by an eighty-residue insert that appears mostly unstructured except for a small β-sheet region, packed against the base of the two helical bundles (Fig. EV5A). Phylogenetic analysis shows that this cryptic domain is broadly distributed in bacteria and present in over 400 proteins with different architectures. In *L. pneumophila* alone, it is found in four effectors (Lpg2527/LnaB, Lpg0437/SidL, Lpg0208, and Lpg0209) (Fig. EV5B). Similar sequences are found in other pathogens, such as *Coxiella burnetii*, several species of *Pseudomonas*, and *Vibrio*, including several strains of *V. cholerae* (Appendix Table S2). Within this predicted domain, we were able to distinguish several highly conserved residues

arranged in a potential catalytic triad (Ser261-His305-Glu309) co-localized between the base of the helical bundle and the β-sheet (Fig. EV5C). The substitution of Ser261 to an alanine residue rescued the yeast toxicity phenotype; however, the expression of the Lpg2527/LnaB His305Arg variant caused toxicity like the wildtype (Fig. 5D,E). The Glu309Ala substitution had a dramatic reduction of protein expressed in yeast cells, thus, resulting in this variant being excluded from our experimental panel. Other conserved residues in the Lpg2527/LnaB model that may be part of this putative active site include Thr263 and Tyr304 (Fig. EV5C); however, individual substitutions of these residues to alanine and arginine, respectively, did not lead to alleviation of this effector's toxicity (Fig. 5E).

# Discussion

Over ten percent of the *L. pneumophila* proteome accounts for effector proteins that are translocated by the Dot/Icm secretion system into the eukaryotic host. Dissecting the individual functions of the largest arsenal of secreted pathogenic factors represents a significant challenge that hampers our understanding of this pneumonia-causing bacterium's infection strategy. The recent dramatic improvements in protein 3D structure prediction have provided valuable new tools for globally assessing the potential functional domain repertoire in *L. pneumophila* effectors. By combining advanced protein 3D structure modeling with a model system phenotypic assay, we present an expansive overview of predicted functional domains in over 360 *L. pneumophila* effectors. Our comprehensive analysis has unveiled a remarkably diverse repertoire of predicted folds and functions. This range of predicted functional domains not only enriches our understanding of *L. pneumophila* pathogenicity, but, more importantly, provides a foundation for the functional character-ization of potential novel functional entities within previously uncharacterized effectors.

Previous large-scale studies of *L. pneumophila* effectors, based on primary sequence analysis, highlighted the presence of a significant number of structural motifs typically present in the protein of eukaryotic organisms, defined as 75% and above of protein sequences being encoded in eukaryotic genomes (Burstein et al, 2016; Gomez-Valero et al, 2019). Among such motifs, so-called tandem repeat motifs, including the ARM, ANK, and LRRs, were identified as the most recurrent in *L. pneumophila* effector proteins. Primarily recognized for their role in protein–protein interactions, these structural elements are now acknowledged as more versatile molecular recognition modules that can also be involved in protein-lipid and protein-sugar interactions (Islam et al, 2018). Confirming the prevalence of tandem repeats in *L. pneumophila* effectors, the analysis of their predicted 3D models suggested an even larger presence of these structural elements, particularly for ARMs. The identification of tandem repeat motifs in conjunction with other predicted functional domains in an effector model may indicate their combined role in interactions with the appropriate host substrate, which, in many cases, awaits identification and functional characterization. Notably, all nine effector models with predicted LRRs lacked other functional determinants. This observation may suggest the unique role of LRRs as autonomous protein regulator molecules in the effector arsenal.

Another eukaryotic-like feature of *L. pneumophila* effectors revealed by our analysis is the overrepresentation of proteins with intrinsically disordered regions (IDRs). IDRs have been increas-ingly recognized as an integral part of a cellular proteome that does not fold into a specific 3D structure, but rather performs their function while maintaining a range of alternative conformations (Wright and Dyson, 1999). IDRs are estimated to be present in over 40% of a given eukaryotic proteome and their role in different cellular processes is only starting to emerge (Latysheva et al, 2015). In the case of analyzed *L. pneumophila* effector models, we estimated IDRs to be present in at least 24% of these predicted effector models, as compared to 4.8% estimated for the remainder of this bacterium's proteome. While these estimates are based on sequence-based IDR predictions, they are further supported or even extended by AlphaFold2 predictions, which are accepted to have high accuracy for the identification of such regions (Zhao et al, 2023). Notably, eight *L. pneumophila* effector models do not contain any recognizable structural elements. While this may be the result of the Alphafold2 algorithm's limitation in predicting structural elements in these effectors, it also raises an intriguing question of IDR effector potential function during infection and possible functional mimicry with host cell IDR counterparts. Such an observation further supports and expands on the previous hypothesis about the evolutionary importance of interkingdom horizontal gene transfer for the acquisition of "eukaryotic-like *L. pneumophila* effectors (de Felipe et al, 2005; Gomez-Valero et al, 2011); therefore, this may also suggest that these effectors may be intrinsically disordered to their full length.

Our analysis also highlighted the prominence of transmembrane helices (TMs) in effector models, thus indicative of membrane localization to host cell organelles or the LCV. Effectors possessing such structural elements remain one of the most understudied categories of the *L. pneumophila* effector arsenal. TMs were identified in 62 of the effector models, with the number of them varying from 1 to 11. Notably, in the case of 22 effectors, TM helices were the only structural element recognized in their model (often accompanied by only disordered regions or helical bundles). In the remaining effectors in this category, models contained other recognizable structural elements or previously unrecognized domains with potential enzymatic activity, such as cysteine protease or α-β hydrolase functionalities.

Our analysis revealed new members of prominent protein families, such as kinases and cysteine proteases, of which two families have been already suggested as having the most representatives in the *L. pneumophila* effector arsenal. For protein kinases, our expands on the list of kinase effectors identified as part of the global survey of *Legionella* kinases (Krysinska et al, 2022) by identifying two additional effectors with potential kinase domains, specifically Lpg1317/RavW and Lpg1684.

In the case of cysteine proteases, we have doubled the estimated number of effectors expected to carry such a domain. These included several effectors with predicted similarity to acetyltrans-ferase domains previously described in YopJ effectors translocated by the T3SS. This highlights the phenomenon of shared host manipulation strategies between bacterial pathogens with drasti-cally diverse pathogenic lifestyles and effector translocation systems. At the other end of protein family prominence, we also identified representatives of protein families that, to our knowledge, have never been associated with effector arsenals, such as osmotin-like domains and the RIFT-related alanine racemase family

(Dataset EV2). The specific roles of such domains in the effectors' function remain to be determined.

Many of the predicted functional domains that we identified in this study were accompanied in effector models by additional structural motifs—such as helical bundles, TMs, or tandem repeats—which may contribute to their cellular function as localization signals or substrate binding determinants. The presence of such additional functional elements could be particularly important for the recruitment of specific substrates in the host, as was shown for the LegK7 kinase where the α-helical elements outside of the main enzymatic domain were shown to be responsible for the recruitment of MOB1A (Lee et al, 2020). Several effector models with cryptic functional domains also included *L. pneumophila* effector-specific substrate recruitment elements. One such example is the helical insertion domain characterized in the aforementioned Lpg2147/MavC and Lpg2148/MvcA effectors to recognize their host targets (Puvar et al, 2020; Valleau et al, 2018). Our analysis suggests the presence of a similar structural element in several effector models, including that of Lpg3000, which is conserved across all sequenced *Legionella* species (Gomez-Valero et al, 2019).

A significant number of structural domains predicted in *L. pneumophila* effector proteins lacked similarity with experimentally characterized protein structures. In 133 of these effectors, the domains consisted of multiple α helices assembled in various α-helical bundles. In 35 effectors, these helical bundles were the only recognized structural motifs. Such structural motifs appear common in a given proteome and are particularly challenging for functional annotation due to the lack of obvious functionally related structural signatures. However, two recently characterized *L. pneumophila* effectors provide an indication of how these structural motifs can be adapted to a specific function in the host cell: Lpg2829/SidH and Lpg2327/Lug15. While the arrangement of the helical bundles in Lpg2829/SidH does not share significant structural similarity with any proteins in the PDB, this atypical structural architecture enables it to bind to human t-RNA (Sharma et al, 2023). Lpg2327/Lug15, despite an exclusively α-helical structure, has been demonstrated to possess E3 ubiquitin ligase activity (Ma et al, 2023). Given the recurrent prediction of unique α-helical structures in *L. pneumophila* effectors, we anticipate more specific activities to be associated with such structural arrangements.

A subset of *L. pneumophila* effector models contained cryptic domain architectures. Notably, some of these proteins shared significant sequence similarity with effectors encoded by other bacterial pathogens suggesting shared functionality. During the revision of this manuscript, recent studies have structurally and functionally characterized Lpg2527/LnaB. Their data not only corroborates our analysis, but also found this effector to be a new structural fold that is part of the ampylase protein family (Fu et al, 2024; Wang et al, 2024). Therefore, effectors with cryptic domains represent an exciting basis for the discovery of new eukaryotic cell manipulation mechanisms employed by *L. pneumophila* (and potentially other bacterial pathogens).

To conclude, by combining the analysis of both primary sequence and 3D structural predictions of all reported *L. pneumophila* effectors with functional assays, we have converged on a full catalog of predicted functions. As a proof-of-concept of our global structural analysis, we precisely pinpoint functional residues that are essential for several cryptic domains by experimentally confirming their importance for the activity of the *L. pneumophila* effector in yeast cells. Many more still await a full experimental verification. By putting together all the annotations

and analyses in the interactive web-based database, we provide a centralized starting point for the functional studies of specific *L. pneumophila* effectors with a clear map of predicted and experimentally characterized functionally relevant 3D elements.

# Methods

**Reagents and tools table**

| Reagent/resource | Reference or source | Identifier or catalog number |
| --- | --- | --- |
| **Experimental models** | | |
| *S. cerevisiae* strain BY4741 | American Type Culture Collection (ATCC) | Cat#201388D-5 |
| **Recombinant DNA** | | |
| pDONR221 | Invitrogen | Cat#12536017 |
| pDONR221-Dot/Icm substrate library | Losick et al, (2010) and Urbanus et al, (2016) | Prof. Dr. Alex Ensminger, University of Toronto, Canada |
| pAG426GAL-FLAG | Urbanus et al, (2016) | Prof. Dr. Alex Ensminger, University of Toronto, Canada |
| pAG426GAL-FLAG-Dot/Icm effectors and their mutants | This study | - |
| pAG416GAL-HA | Urbanus et al, (2016) | Prof. Dr. Alex Ensminger, University of Toronto, Canada |
| pAG416GAL-HA-Dot/Icm effectors and their mutants | This study | - |
| **Antibodies** | | |
| Mouse anti-FLAG | Cell Signaling Technology | Cat#8146S |
| Mouse anti-GAPDH | Cell Signaling Technology | Cat#97166S |
| Goat anti-Rabbit IgG | Abcam | Cat#ab6721 |
| Rabbit anti-HA | Cell Signaling Technology | Cat#3724S |
| Rabbit anti-Goat IgG | Abcam | Cat#ab6721 |
| **Oligonucleotides and other sequence-based reagents** | | |
| QuikChange Primers | This study | Appendix Tables 3 and 4 |
| **Chemicals, Enzymes and other reagents** | | |
| Ampicillin | BioShop | Cat#AMP201.100 |
| Dextrose | Fisher Scientific | Cat#D16-10 |
| Galactose | BioShop | Cat#GAL500.500 |
| Gateway BP Clonase II Enzyme Mix | Thermo Fisher Scientific | Cat#11789100 |
| Gateway LR Clonase II Enzyme Mix | Thermo Fisher Scientific | Cat#11791020 |
| Immobilon Western chemiluminescent HRP substrate | Millipore Sigma | Cat#WBKLS0500 |
| Kanamycin | BioShop | Cat#KAN201.50 |
| PRESTO miniprep plasmid extraction kit | Geneaid | PDH100 |

| Reagent/resource | Reference or source | Identifier or catalog number |
|---|---|---|
| Nitrocellulose Membrane | Bio-Rad | Cat#1620115 |
| Phusion High-Fidelity DNA Polymerase Kit | New England Biolabs | M0530S |
| **Software** | | |
| AlphaFold | https://github.com/google-deepmind/alphafold | - |
| BlastP | https://blast.ncbi.nlm.nih.gov/Blast.cgi | - |
| ChimeraX | https://www.cgl.ucsf.edu/chimerax/ | - |
| Consurf | https://consurf.tau.ac.il/consurf_index.php | - |
| EMBOSS Needle | https://www.ebi.ac.uk/jdispatcher/psa/emboss_needle | - |
| Evolutionary-scale prediction Metagenonomic Atlas | https://esmatlas.com/resources?action=fold | - |
| FATCAT | https://fatcat.godziklab.org/ | - |
| Foldseek | https://search.foldseek.com/search | - |
| HHPred | https://toolkit.tuebingen.mpg.de/tools/hhpred | - |
| **Other** | | |
| ChemiDoc Touch Gel Imaging System | Bio-Rad | - |

## 3D modeling of *L. pneumophila* effectors and assignment of domain types based on ECOD hierarchy

Primary sequences and 3D structural models of all 368 *L. pneumophila* effector proteins were analyzed to assign their domain architectures and, when possible, predict potential biochemical functions.

The 3D models calculated with AlphaFold v2.0 (Jumper et al, 2021) were retrieved from the AlphaFold DB database (Varadi et al, 2022), where available or built locally. For the six longest effectors (Lpg0090, Lpg0693, Lpg2153, Lpg2156, Lpg2239, and Lpg2490), which exceeded AlphaFold2 limits, sequences were split into overlapping fragments, which were then modeled individually. In the first step, effector regions were assigned to ECOD domains based on sequence similarity where possible (i.e., BlastP alignment). If the remaining sequence fragments still exceeded the limits of AlphaFold2, they were divided into overlapping fragments of 500 or less residues. The fragments were used to build 3D models. In cases where the initial fragment boundaries appeared to fall within structural domains, the fragment boundaries were revised, and the modeling was repeated. This process was repeated until models of intact domains were built for each fragment. In cases when

AlphaFold2 models had low significance due to lack of sufficient depth of the multiple sequence alignment, we used EMSfold, an LLM-based algorithm that does not rely on multiple sequence alignment for the prediction (Lin et al, 2023).

Next, unstructured regions of the 3D models were removed, and the structured regions were divided into compact fragments corresponding to potential structural domains. These putative domains were then compared to structures of domains of experimentally characterized proteins retrieved from the ECOD database version 288 (Cheng et al, 2014) and reduced by clustering by sequence identity with a 40% cut-off. The structural comparison was performed using the FATCAT program (Ye and Godzik, 2003) without allowing for twists in the aligned structures. The results of FATCAT searches were analyzed, and the putative domains were assigned to domain types using the ECOD topology (T) level when possible (132 domain types are assigned at this level). In cases when structural similarities did not allow for clear assignment at the topology level, the assignment was done at the higher ECOD homology (H) level (19 domain types are assigned at this level). In a few cases, strong sequence and/or structural similarity made it possible to assign effector domains to ECOD at the family level (F) (six of the assigned domain types correspond to ECOD families). In cases when, according to the FATCAT results, the fragments of effector models did not correspond to complete ECOD domains, the initial division of models into potential domains was revised, and FATCAT searches were repeated. For cases where FATCAT didn't find any statistically significant structural matches in the PDB or ECOD databases, we also utilized the DALI server (Holm, 2022), but in all cases, no additional similarities were found.

Independently, the primary sequences of the effectors were compared to primary sequences of entries from the ECOD and PDB databases using BlastP (Altschul et al, 1997) and HHPred (Gabler et al, 2020) algorithms. The primary sequence similarities, if significant, were used to verify structure-based assignments of regions to the ECOD domain topologies and to resolve some cases where assignments to ECOD topologies were difficult to make based on structural similarity. The data about the presence and locations of signal peptides and transmembrane helices were downloaded from the UniProt database (https://www.uniprot.org). Regions with at least two consecutive transmembrane helices were labeled as "transmembrane regions" and included in the annotations of domain architectures unless it was possible to assign them to specific ECOD domain types. Long stretches of structural disorder regions were indicated by the lack of structured domains as predicted with AlphaFold, were classified as "disordered regions" and also included in the descriptions of domain architectures.

In addition, AlphaFold2 models were compared to a large library for predicted protein structures using the FoldSeek server (van Kempen et al, 2024). While FoldSeek recognition of structural similarity to the structures from the PDB lagged that of FATCAT or DALI, FoldSeek matches were used to verify and analyze the structural level distant homology predictions obtained by HHPred, as described in the previous paragraph.

In over 120 effector proteins, we found regions modeled by AlphaFold as a series of helical hairpins or bundles without unambiguous matches to structurally characterized proteins. If their primary sequences also did not show any similarity to experimentally characterized structures, we labeled these fragments as "helical regions".

## Identification of eukaryotic-like domains in *L. pneumophila* effectors

Microbial protein families and domains with mostly eukaryotic homologs are regarded as eukaryotic-like domains. They are more likely to be involved in interference with the signaling and metabolism of the host's cell. Here, the eukaryotic-like domains were identified among effector domains as follows:

1. The sequences of ECOD representatives of a given domain (clustered by sequence identity with a 40% cut-off) were used to start BlastP searches against the set of 22,925 representative proteomes downloaded from the UniProt database (https://www.uniprot.org).
2. The percentage of significant (e-values <0.001) unique BlastP hits (presumed homologs) which came from eukaryotic organisms was calculated for each domain type.
3. Domain types with more than 75% of eukaryotic homologs were labeled as eukaryotic-like.

According to this criterion, we labeled 29 out of the 153 identified ECOD domain types as eukaryotic-like (Dataset EV2).

## Identification of *L. pneumophila* effector domains present in effectors from other species

The effector domains with homologs in other species were labeled as follows:

1. The sequences of all ECOD representatives of a given domain type (clustered by sequence identity with a 40% cut-off) were used to start BlastP searches against the SecretEPDB database (An et al, 2017) (the *L. pneumophila* effectors themselves were excluded from the set of SecretEPDB sequences).
2. The effector domains with significant (e-values <0.001) BlastP hits in the SecretEPDB database were labeled as being present in effectors from species other than *L. pneumophila*.

With the above procedure, we labeled 46 out of the 153 identified domain types as present in known effectors from other organisms.

## Cloning and DNA manipulations

The pDONR221-effector constructs used in this study were obtained from the pDONR221-Dot/Icm substrate library from a previous report (Losick et al, 2010; Urbanus et al, 2016). Point mutations of each wildtype effector in pDONR221 were introduced using QuikChange, as previously described (Liu and Naismith, 2008). Primers used for the generation of point mutations can be found in Appendix Tables S3 and S4. The plasmid DNA of each construct was obtained using a PRESTO miniprep plasmid extraction kit and validated by sequencing. All pDONR221 constructs containing the wild-type effectors and their variants were then cloned into the high-copy vector pAG426GAL-FLAG-ccdB, or the low-copy vector pAG416GAL-HA-ccdB (Urbanus et al, 2016) (Lpg2461 and Lpg2523/Lem26), using the Gateway LR clonase kit.

## Yeast spotting assays

All the overexpression constructs were transformed into the *S. cerevisiae* BY4741 strain using a previously described lithium acetate procedure (Salomon and Sessa, 2010). The spotting assays of all toxic effectors and their respective variants in this study were performed as previously described with minor modifications (Salomon and Sessa, 2010). In brief, cultures of yeast-carrying vectors with toxic effectors and their mutants were grown in a synthetically defined, selective medium lacking uracil (SD-Ura) and supplemented with 2% dextrose overnight. Following overnight growth, cultures were normalized to an $OD_{600}$ of 1 and used to make tenfold dilutions. Each effector and its mutants were spotted on an SD media plate lacking uracil containing either 2% dextrose (non-inducing) or 2% galactose (inducing). After spotting, plates were incubated for three days at 30 °C and then imaged. All spotting assays were performed in triplicate.

## Western blot analysis to validate effector expression in yeast

The expression and solubility of the toxic effectors and their mutants were performed using western blot analysis with a previously described methodology where only a few modifications were made (Salomon and Sessa, 2010; Urbanus et al, 2016). Overnight cultures (3 ml) in selective SD-Ura and supplemented with 2% dextrose were spun down and washed three times with ddH2O. Cultures were diluted to an $OD_{600}$ of 1 in SD-Ura supplemented with 2% dextrose or 2% galactose, and then grown overnight. The overnight cultures were spun down, and the yeast cell pellets were resuspended in 100 µl of ice-cold lysis buffer (4% v/v 5 M NaOH and 0.5% v/v β-mercaptoethanol) and incubated on ice for 40 min. After incubation, 1 µl of 6 N HCl and 50 µl of 3x sample loading buffer (0.05% v/v bromophenol blue, 30% v/v glycerol, 37.5% v/v 500 mM Tris-HCl pH 6.8, 0.15% w/v sodium dodecyl sulfate [SDS], and 500 mM Dithiothreitol) was added to the samples and mixed. Samples were then boiled at 95 °C for 5 min and centrifuged for 1 min at $12000 \times g$. About 20 µl of the sample was analyzed on a 12% SDS-PAGE gel for immunoblot analysis. Proteins were transferred onto a nitrocellulose membrane and then blocked with 5% non-fat milk in tris-buffered saline-0.1% Tween 20 detergent (TBST) buffer for 2 h at room temperature (RT). After blocking, the membrane was then washed three times for 15 min with tris-buffered saline-0.1% Tween 20 detergent (TBST) buffer, and then incubated overnight at 4 °C with a primary antibody that was specific for the N-terminal fusion tag of each overexpression construct (α-FLAG or α-HA) at a dilution of 1:1000 in 5% non-fat milk TBST. Following primary antibody incubation, the membrane was washed with TBST three times for 15 min. After the washing, the membranes were incubated with the appropriate horse radish peroxidase-conjugated secondary antibody (Rabbit to mouse IgG or Goat to rabbit IgG) for 1 h at RT. The signal was detected with an Immobilon Western chemiluminescent HRP substrate. Immunoblots were visualized using a Bio-Rad ChemiDoc machine. The loading control western blots were performed with an α-GADPH antibody at 1:1000 dilution in a 5% non-fat milk TBST solution. The GADPH westerns were performed as the previously mentioned procedure.

A blinding was not used for this study.

# Data availability

Analysis of all effector models can be viewed in the following database: https://pathogens3d.org/legionella-pneumophila. Request for reagents and plasmids will be fulfilled upon request by the corresponding author Dr. Alexei Savchenko (alexei.savchenko@ucalgary.ca).

The source data of this paper are collected in the following database record: biostudies:S-SCDT-10_1038-S44320-024-00076-z.

# Peer review information

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

## Acknowledgements

We would like to thank Dr. Nobuhiko Watanabe and Dr. Elina Karimullina from the Savchenko Laboratory for their insightful discussions. We would also like to thank Dr. Beth Nicholson for their assistance with cloning. The work from this manuscript was funded in whole or in part with U.S. Federal funds from the National Institute of Allergy and Infectious Diseases, National Institutes of Health Department of Health and Human Services, which is under Contract numbers: HHSN272201700060C and 75N93022C00035. These funds were awarded to the Center for Structural Biology of Infectious Diseases (CSBID, http://csbid.org). The functional data in this manuscript was supported by a Canadian Institutes of Health Research Project Grant (AE and AS) (PJT-162256) and a National Science and Engineering Research Council of Canada (NSERC) Discovery program grant (RGPIN-2017-04878) that was awarded to the principal investigator, AS. Computations were performed using the computer clusters and data storage resources of the UCR High-Performance Computer Cluster (HPCC), which were funded by grants from NSF (MRI-2215705, MRI-1429826) and NIH (1S10OD016290-01 A1). DTP was supported by the Doctoral Alberta Graduate Excellence Scholarship.

## Author contributions

**Deepak T Patel**: Data curation; Formal analysis; Validation; Investigation; Visualization; Methodology; Writing—original draft; Writing—review and editing. **Peter J Stogios**: Data curation; Formal analysis; Validation; Investigation; Visualization; Methodology; Writing—review and editing. **Lukasz Jaroszewski**: Resources; Data curation; Software; Formal analysis; Validation; Investigation; Visualization; Methodology; Writing—review and editing. **Malene L Urbanus**: Data curation; Formal analysis; Validation; Writing—review and editing. **Mayya Sedova**: Resources; Data curation; Software. **Cameron Semper**: Data curation; Formal analysis; Validation; Methodology. **Cathy Le**: Data curation; Formal analysis; Investigation; Visualization; Methodology; Writing—review and editing. **Abraham Takkouche**: Data curation; Software; Methodology. **Keita Ichii**: Data curation; Software; Methodology. **Julie Innabi**: Data curation; Software; Methodology. **Dhruvin H Patel**: Data curation; Formal analysis; Investigation. **Alexander W Ensminger**: Conceptualization; Supervision; Funding acquisition; Visualization; Project administration; Writing—review and editing. **Adam Godzik**: Conceptualization; Data curation; Software; Formal analysis; Supervision; Funding acquisition; Validation; Investigation; Visualization; Methodology; Project administration; Writing—review and editing. **Alexei Savchenko**: Conceptualization; Resources; Data curation; Formal analysis; Supervision; Funding acquisition; Investigation; Visualization; Methodology; Writing—original draft; Project administration; Writing—review and editing.

Source data underlying figure panels in this paper may have individual authorship assigned. Where available, figure panel/source data authorship is listed in the following database record: biostudies:S-SCDT-10_1038-S44320-024-00076-z.

## Disclosure and competing interests statement

The authors declare no competing interests.

# Expanded View Figures

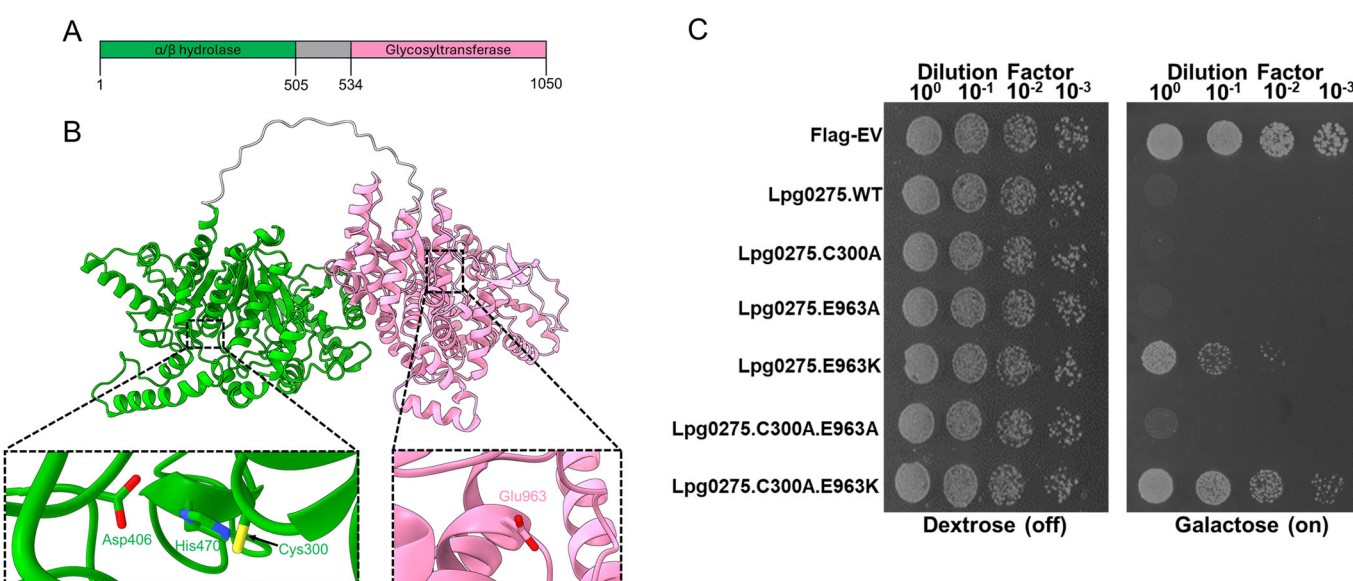

**Figure EV1.   Both domains of Lpg0275/SdbA are involved in yeast toxicity.**

(**A**) Schematic of the domain organization of Lpg0275/SdbA. The N-terminal domain shown in green corresponds to the hydrolase domain, whereas the pink indicates the glycosyltransferase domain. (**B**) Alphafold2 model of Lpg0275/SdbA. The green represents the hydrolase domain, and the pink represents the glycosyltransferase domain. Below is a zoomed-in view of the predicted catalytic residues of each predicted enzymatic domain. (**C**) Yeast toxicity panel of strains expressing FLAG-tagged constructs of full-length wildtype Lpg0275 and its variants.

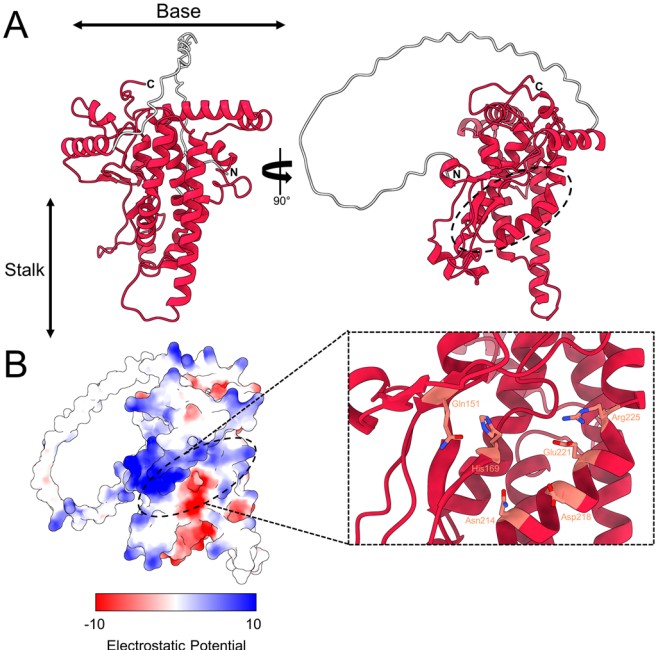

**Figure EV2.  Lpg1154/RavQ forms a unique "T" shape containing a highly conserved groove that may serve as an active site.**

(**A**) The Lpg1154/RavQ model (residues 59–389, red) shows the base and stalk that form the "T" shape. (**B**) An electrostatic potential surface representation of a potential active site cavity of Lpg1154/RavQ, followed by a zoom-in of the conserved residues identified in Lpg1154/RavQ and its orthologs from the *Legionella* genus, which are arranged in a potential active site.

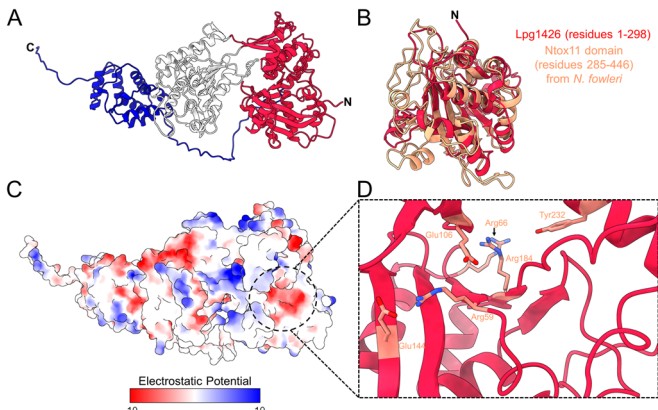

**Figure EV3. Lpg1426/VpdC has a cryptic domain on the N-terminus that has structural similarity to the Ntox11 putative toxin found in human pathogenic amoeba.**

(A) Cartoon representation of the Lpg1426/VpdC Alphafold2 model. The cryptic domain is found on the N-terminus (red), followed by a central phospholipase domain (white) and the C-terminal helical bundle involved in interactions with ubiquitin. (B) Structural alignment of the Lpg1426/VpdC cryptic domain (residues 1–298, red) onto the Ntox11 Alphafold2 model (residues 285–446, salmon) from *N. fowleri*. (C) Surface representation of the electrostatic potential of the Lpg1426/VpdC model that also shows a conserved negatively-charged pocket (dotted circle) is present in the cryptic domain. (D) Zoom in on the positively charged region where highly conserved residues (salmon sticks), which are present in the *Legionella* orthologs of Lpg1426/VpdC, form a pocket.

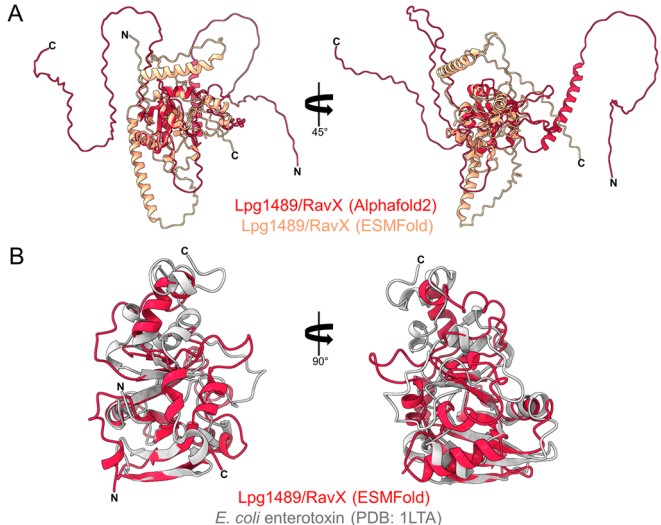

A

Lpg1489/RavX (Alphafold2)
Lpg1489/RavX (ESMFold)

B

Lpg1489/RavX (ESMFold)
*E. coli* enterotoxin (PDB: 1LTA)

**Figure EV4.   Lpg1489/RavX has a central globular cryptic domain surrounded by disordered loops.**

(A) Structural alignment of the full-length Lpg1489/RavX models generated by Alphafold2 (red) and ESMFold (salmon). (B) ESMFold model of the Lpg1489/RavX cryptic domain (residues 83–263, red) onto the *E. coli* enterotoxin (PDB: 1LTA, residues 1–181, gray).

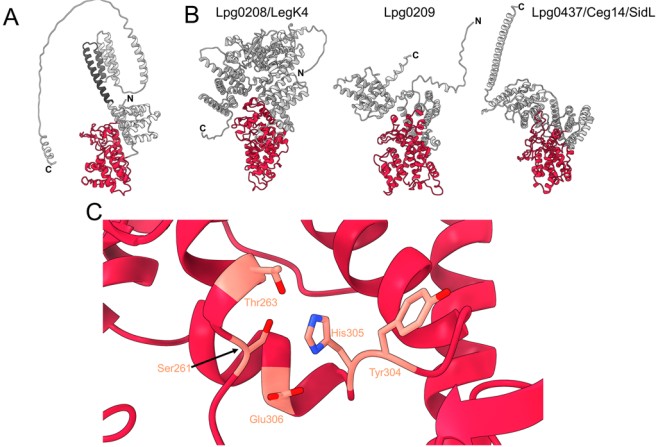

**Figure EV5. The cryptic domain of Lpg2527/LnaB is present in other *L. pneumophila* effectors, harboring a conserved set of residues that resemble a potential active site.**

(A) Alphafold2 model of Lpg2527/LnaB which highlights the cryptic domain (red) and the helical bundle that was previously shown to be important in the activation of the NF-κB pathway (dark gray) (Losick et al, 2010). (B) Representation of the Lpg2527/LnaB cryptic domain (shown in red) that is present in other *L. pneumophila* effectors (Lpg0208/LegK4, Lpg0209, and Lpg0437/Ceg14/SidL). (C) Zoomed-in image of the putative active site of Lpg2527/LnaB that is also conserved in its *Legionella* orthologs and other *L. pneumophila* effectors containing this domain.

