## [Peer Review File · Molecular Systems Biology]

Global atlas of predicted functional domains in *Legionella pneumophila* Dot/Icm translocated effectors

Deepak Patel, Peter Stogios, Lukasz Jaroszewski, Malene Urbanus, Mayya Sedova, Cameron Semper, Cathy Le, Abraham Takkouche, Keita Ichii, Julie Innabi, Dhruvin Patel, Alexander Ensminger, Adam Godzik, and Alexei Savchenko

Corresponding author(s): Alexei Savchenko (alexei.savchenko@ucalgary.ca) , Alexander Ensminger (alex.ensminger@utoronto.ca), Adam Godzik (adam.godzik@medsch.ucr.edu)

Review Timeline:

Submission Date:	21st May 24
Editorial Decision:	3rd Jul 24
Revision Received:	21st Aug 24
Editorial Decision:	27th Sep 24
Revision Received:	17th Oct 24
Accepted:	31st Oct 24

Editor: Poonam Bheda

Transaction Report:

3rd Jul 2024

Manuscript Number: MSB-2024-12436-T

Title: Global atlas of predicted functional domains in *Legionella pneumophila* Dot/Icm translocated effectors

Dear Dr. Savchenko,

Thank you for the submission of your manuscript to Molecular Systems Biology. We have now received feedback from the three reviewers who agreed to evaluate your manuscript. As you will see from the reports below, the referees acknowledge the interest of the study and are overall supporting publication of your work pending appropriate revisions.

I think that the recommendations of the reviewers are rather clear and I therefore do not see the need to repeat the comments listed below. One of the more fundamental points raised by both Reviewers 1 and 3 refers to the organization of the results, which seem as a long series of observations, and could benefit from some reformatting to make it more accessible to readers.

All other issues raised would need to be satisfactorily addressed. Please let me know in case you would like to discuss in further detail any of the issues raised, I would be happy to schedule a call.

We require:

4) A .docx formatted letter INCLUDING the reviewers' reports and your detailed point-by-point responses to their comments. As part of the EMBO Press transparent editorial process, the point-by-point response is part of the Peer Review File (PRF), which will be published alongside your paper.

5) A complete author checklist, which you can download from our author guidelines (<https://www.embopress.org/page/journal/17574684/authorguide#submissionofrevisions>). Please insert information in the checklist that is also reflected in the manuscript. The completed author checklist will also be part of the PRF.

6) Please note that all corresponding authors are required to supply an ORCID ID for their name upon submission of a revised manuscript.

7) It is mandatory to include a 'Data Availability' section after the Materials and Methods. Before submitting your revision, primary datasets produced in this study need to be deposited in an appropriate public database, and the accession numbers and database listed under 'Data Availability'. Please remember to provide a reviewer password if the datasets are not yet public (see <https://www.embopress.org/page/journal/17574684/authorguide#dataavailability>).

In case you have no data that requires deposition in a public database, please state so in this section. Note that the Data Availability Section is restricted to new primary data that are part of this study. This study includes no data deposited in external repositories.

8) For data quantification: please specify the name of the statistical test used to generate error bars and P values, the number (n) of independent experiments (specify technical or biological replicates) underlying each data point and the test used to calculate p-values in each figure legend. The figure legends should contain a basic description of n, P and the test applied. Graphs must include a description of the bars and the error bars (s.d., s.e.m.). Please provide exact p values.

9) Our journal encourages inclusion of *data citations in the reference list* to directly cite datasets that were re-used and obtained from public databases. Data citations in the article text are distinct from normal bibliographical citations and should directly link to the database records from which the data can be accessed. In the main text, data citations are formatted as follows: "Data ref: Smith et al, 2001" or "Data ref: NCBI Sequence Read Archive PRJNA342805, 2017". In the Reference list,

data citations must be labeled with "[DATASET]". A data reference must provide the database name, accession number/identifiers and a resolvable link to the landing page from which the data can be accessed at the end of the reference. Further instructions are available at .

<https://www.embopress.org/page/journal/17574684/authorguide#expandedview>

11) For more information: There is space at the end of each article to list relevant web links for further consultation by our readers. Could you identify some relevant ones and provide such information as well? Some examples are patient associations, relevant databases, OMIM/proteins/genes links, author's websites, etc...

12) Author contributions: CRediT has replaced the traditional author contributions section because it offers a systematic machine readable author contributions format that allows for more effective research assessment. Please remove the Authors Contributions from the manuscript and use the free text boxes beneath each contributing author's name in our system to add specific details on the author's contribution. More information is available in our guide to authors.

13) Disclosure statement and competing interests: We updated our journal's competing interests policy in January 2022 and request authors to consider both actual and perceived competing interests. Please review the policy <https://www.embopress.org/competing-interests> and update your competing interests if necessary.

14) Every published paper now includes a 'Synopsis' to further enhance discoverability. Synopses are displayed on the journal webpage and are freely accessible to all readers. They include a short stand first (maximum of 300 characters, including space) as well as 2-5 one-sentences bullet points that summarizes the paper. Please write the bullet points to summarize the key NEW findings. They should be designed to be complementary to the abstract - i.e. not repeat the same text. We encourage inclusion of key acronyms and quantitative information (maximum of 30 words / bullet point). Please use the passive voice. Please attach these in a separate file or send them by email, we will incorporate them accordingly.

Please also suggest a striking image or visual abstract to illustrate your article as a PNG file 550 px wide x 300-600 px high. Share synopsis text and image, as well as eTOC:

Please note that these would be the final versions and changes during proofing are usually not allowed

15) As part of the EMBO Publications transparent editorial process initiative (see our Editorial at <http://embomolmed.embopress.org/content/2/9/329>), Molecular Systems Biology Medicine will publish online a Peer Review File (PRF) to accompany accepted manuscripts.

In the event of acceptance, this file will be published in conjunction with your paper and will include the anonymous referee reports, your point-by-point response and all pertinent correspondence relating to the manuscript. Let us know whether you agree with the publication of the PRF and as here, if you want to remove or not any figures from it prior to publication.

Please note that the Authors checklist will be published at the end of the PRF.

Molecular Systems Biology has a "scooping protection" policy, whereby similar findings that are published by others during review or revision are not a criterion for rejection. Should you decide to submit a revised version, I do ask that you get in touch after three months if you have not completed it, to update us on the status.

I look forward to receiving your revised manuscript.

Yours sincerely,

Poonam Bheda, PhD
Scientific Editor
Molecular Systems Biology

Reviewer #1:

Patel et al. conducted a global analysis of structural models generated mostly by AlphaFold2 of 368 *L. pneumophila* effectors. They divided the effectors into different enzyme families and analyzed them for the presence of globular domains and structural motifs. Some predictions were validated via ectopic expression of effectors in the *Saccharomyces cerevisiae* model system. They also identified 30 effectors containing no structural similarity to any experimentally characterized 3D structure. There is no sophisticated new method in the paper as far as I can see but it provides a valuable atlas for effector functions in *Legionella*.

For the authors, to make future papers more readable and easier to review:

1. Please provide page and line numbers.
2. No figure numbers were provided in the merged file.

Major Revisions:

1. The description of the biochemical activities of some effector proteins only based on structural similarity is way too long. I would shorten this analysis by at least 50% to make the manuscript more readable and interesting. Currently the text of this analysis stretches for 8 pages. Figures 2-6 are nearly duplications of each other, just for different protein groups. All show structural alignment and zooming into the catalytic sites. If space was limited these could have been shrunk in my view to fewer main figures (combining multiple families) and more supplementary figures.
2. To make the bioinformatic analysis more interesting I suggest to divide the results shown in Fig. 9 and attach the panels to the relevant figures showing the computational analysis of each domain. This way the reader will get an impression of computational prediction directly followed by experimental validation.
3. Newly released tools should be employed in the analysis including Foldseek which should be better than Dali for structural comparison and AlphaFold3 which can be used to predict certain ligands and substrates of proteins, including some of the effector proteins discussed in this work.
4. "The Cys300Ala substitution of the α/β hydrolase domain significantly reduced the toxicity compared to the wild-type expression" - in Fig. 9A it seems to be highly toxic.

Minor Revisions:

1. The sentence: "Our analysis identified 42 distinct domain types in 64 *L. pneumophila* effectors causing toxicity in the yeast model system". There is zero description in the results section of the experimental system used to express the genes in yeast in this part. Please provide this description for better understanding. Alternatively remove this sentence or mention that it will be described later in the manuscript.
2. Table 1 headers and cells are trimmed and cannot be fully read. It would be best not to convert the table into a PDF but to keep it as a spreadsheet, or at least make sure that it is fully readable before submission.
3. "The toxicity of this effector to yeast was linked to its cysteine protease domain and was shown to be alleviated by Cys280Ser" - the figure 9A shows C280A.

Reviewer #2:

SUMMARY

This study seeks to broaden understanding of the structure and function of *Legionella pneumophila* effector proteins. With a hefty number of 368 previously known effectors, identifying and characterizing structural domains presents both a challenge and an opportunity. Through an approach that combined existing structural data, AI-built structural models and yeast functional assays, the work detailed in this manuscript provides a comprehensive, careful analysis of predicted functional domains among effectors. A number of new and important insights emerged from this work, including:

- Revealing cysteine proteases as the most prevalent, yet functionally diverse class of *Legionella* effectors. Notably, analyses more than doubled the number of cysteine proteases known prior to this work.
- Expanding the number of effectors that harbor tandem repeat motifs typically found in eukaryotic organisms and known to be functionally versatile.
- The observation that IDRs are more abundant among effectors than the rest of the proteome.

- Predicting new functional domains including coordinates of their catalytic motifs. These are valuable data that can inform future in-depth studies.
- Revealing the prevalence of effectors harboring transmembrane helices.
- The identification of numerous cryptic domains, some predicted to have functions that have been associated with Legionella effectors and others predicted to have new functions.
- Identifying a subset of effectors with new, unique folds, some being predicted to share functions with effectors from other species. Probing the functionality of these unique folds through yeast growth assays which in three of the four tested effectors led to identification of critical residues. Even finding that many other domains with predicted novel folds were not lethal is useful information for future studies.
- Creating a publicly available interactive web-based database, providing immediate and unrestricted access to new, valuable data that can inform future functional studies.

MAJOR POINTS

None

MINOR POINTS

- For those effectors that harbor IDRs, is there an indication of whether these regions tend to be closer to the N- or C-terminus of the protein, or elsewhere in the protein.
- In terms of the distribution of residues across the 368 effectors, are cysteines more prevalent in effector proteins than the rest of the proteome?
- In many protozoa and organisms that harbor cysteine proteases, their activity can be regulated by cysteine protease inhibitors such as cystatins. Is there an indication of whether the Legionella effectorome (or genome) harbors proteins that function as cysteine protease inhibitors?

CONCLUSION

Overall, this is a very well-written manuscript describing an elegant, thorough, and timely study. It was a pleasure to read. The clever use of advanced computational tools and traditional methods resulted in a significant expansion and refinement of current knowledge of the structure and function of Legionella effectors. In my opinion, these contributions are more than sufficient to warrant publication of this manuscript as is. The wealth of new information made available here will positively impact future studies in Legionella pathogenesis and more broadly in the host-microbe interaction field.

Reviewer #3:

The study by Patel et al., discovers a large number of functional domains and unique domains in an array of legionella effectors. They also go onto identify catalytic residues in a number of these domains and show how they affect yeast toxicity (a common functional assay for legionella effectors). This is a great resource for legionella effector community in general and possibly also for researchers exploring effectors of other pathogenic bacteria as many of these newly identified domains seem to be also present in other pathogens. My specific comments are listed below.

Major comment:

- 1) The paper reads more like a book with too many names and results presented as a matter of fact. It would be nice for eg. to briefly mention if an effector's role in intracellular replication has been studied previously or not for each effector discussed. Or this information can also be in a table. Without this context, much of the manuscript reads like a series of observations with no biological implications immediately apparent.
- 2) It would help the papers impact if the authors can verify at least a couple of biochemical activities predicted by AF. I understand the lack of information on substrates is a roadblock but some enzymes (eg. kinases) self-modify so this can be used to verify their activity relatively easily. Although yeast toxicity experiments are informative, biochemical assays are required to validate the domain predictions of AF.

Minor comments.

In the first section of results, authors write that "This level of confidence was shown to correspond to 100% correct fold assignments in numerous independent experiments, including those performed by our groups." It would be nice to support such

statements with appropriate references.

In the server link, overall, the database is nicely organized. But I found that when I tried to download some PDB files, the server just did not respond. Can the authors look into this? I randomly looked up a few effectors and Lpg2153 and Lpg0376 AF PDBs were not available.

SidJ should be mentioned and discussed in the section about LP kinases.

Figure 1: It would be easier if the authors can include what each color means within the figure itself. Also authors should mention the specifics of what they mean by remote homology and close homology in the figure legend.

In the "Cryptic domain activity..." section, at the very end, the authors talk about the Lpg2523/Lem26 PDE mutant still showing toxicity in yeast.

I was a little confused by the term cryptic domains. This usually means that the domain's function is unknown. This term "cryptic" is more appropriate for "unique" domains. Instead of cryptic, the authors should use the term "predicted functional domain" especially in the title of the section. In fact, authors used this term later in the description to refer to these exact domains.

In the final section of results, the authors use ESM Fold instead of AF2. I am curious as to how the model accuracy generated using ESM fold can be compared to the AF2 model accuracy to judge which is better. Can the authors comment if they observe similar catalytic residues using both the predicted models?

In the final section of discussion, authors state "To conclude, by combining advanced 3D structural predictions of all reported *L.pneumophila* effectors with functional assays, we have converged on a full catalog of predicted functions and mechanisms of how this pathogen takes control and thrives inside eukaryotic phagocytic cells." I totally agree with the first part of this sentence but the manuscript does not shed light on how pathogen takes control and thrives inside phagocytic cells. This needs to be removed or reworded to accurately reflect the findings.

Table 1: the table legend is not detailed enough to understand all the details of the table contents. Each column heading should be explained in the legend.

In the methods, 3D modeling section, authors seem to have split sequences for AF prediction for select effectors. These need to be mentioned in the section.

***Replies to reviewers are in red ***

Reviewer #1:

Patel et al. conducted a global analysis of structural models generated mostly by AlphaFold2 of 368 *L. pneumophila* effectors. They divided the effectors into different enzyme families and analyzed them for the presence of globular domains and structural motifs. Some predictions were validated via ectopic expression of effectors in the *Saccharomyces cerevisiae* model system. They also identified 30 effectors containing no structural similarity to any experimentally characterized 3D structure. There is no sophisticated new method in the paper as far as I can see but it provides a valuable atlas for effector functions in *Legionella*.

We thank the reviewer for taking the time to read and critically assess our manuscript, while also appreciating the value of this resource to the field.

For the authors, to make future papers more readable and easier to review:

1. Please provide page and line numbers.

Page and line numbers are provided in revised manuscript.

2. No figure numbers were provided in the merged file.

We have also addressed this issue in the revised manuscript.

Major Revisions:

1. The description of the biochemical activities of some effector proteins only based on structural similarity is way too long. I would shorten this analysis by at least 50% to make the manuscript more readable and interesting. Currently the text of this analysis stretches for 8 pages. Figures 2-6 are nearly duplications of each other, just for different protein groups. All show structural alignment and zooming into the catalytic sites. If space was limited these could have been shrunk in my view to fewer main figures (combining multiple families) and more supplementary figures.

We have done our best to reduce the text of the manuscript to help with the readability. The Figures have been rearranged to focus on each of the predicted domain group and include the specific representatives that have been chosen for analysis in yeast toxicity assay. Furthermore, we have trimmed the manuscript to aid in the readability of the manuscript as can be viewed from extensive changes visualised in the track changes version of the draft. We have reduced sentences in multiple ways; for instance, removing long transitions, or removed general statements from specific sections of the draft that, in our opinion, did not contribute significantly to the understanding of structural analysis presented in the draft. For example, we removed the section describing the analysis and previous knowledge of LicA effector.

2. To make the bioinformatic analysis more interesting I suggest to divide the results

shown in Fig. 9 and attach the panels to the relevant figures showing the computational analysis of each domain. This way the reader will get an impression of computational prediction directly followed by experimental validation.

We agree to the reviewer's suggestion and have changed the Figures to include the yeast toxicity panels along with structural analysis of the models. Figures illustrating the structural analysis of effector models that do not cause toxicity in yeast have been shifted to the supplementary section of this manuscript. Reflective to this change, we have updated the figure captions in the main text and added additional figures to the "supplementary material" file.

3. Newly released tools should be employed in the analysis including Foldseek which should be better than Dali for structural comparison and AlphaFold3 which can be used to predict certain ligands and substrates of proteins, including some of the effector proteins discussed in this work.

In response to this suggestion, we would like to remind that AlphaFold3 (AF3) was released in May 2024, after all the analysis presented in the paper was completed. With its availability limited to 20 jobs a day, we decided that it would be unrealistic to repeat all the calculations for the paper using AF3. Moreover, tests performed by our group and others suggest that AF3 improvement over AF2 in the accuracy of single protein predictions – an approach we used in our paper, is minimal, as most of the novelty in AF3 lays in its ability to include cofactors, small molecule ligands and build models of large, multi-protein complexes.

Regarding the FoldSeek algorithm it is indeed faster, but less accurate than Dali or FATCAT algorithms. In comparing its results in searches against PDB, FoldSeek recovered about 50% of the matches obtained with Dali or FATCAT. Therefore, for searches against the PDB database we used FATCAT and, when this failed, the Dali algorithm. The main advantage of FoldSeek is in its ability to search large databases of AF models and thus find a potential structural similarity between two proteins with no experimental structures. We have used it extensively in this capacity, in particular running all novel folds against databases of AF predictions. Many distant homologies matches between (potential) novel folds discussed in the paper were obtained by FoldSeek. This was described in the methods section, but we now expanded this description.

4. "The Cys300Ala substitution of the α/β hydrolase domain significantly reduced the toxicity compared to the wild-type expression" - in Fig. 9A it seems to be highly toxic.

We thank the reviewer for pointing out this discrepancy. Indeed, the mutation does not completely alleviate the toxicity; however, in Figure 3, we observe that there is a slight alleviation of toxicity in case of Cys300Ala variant. We have edited the text of the manuscript to reflect these data more accurately (see lines 631-633).

Minor Revisions:

1. The sentence: "Our analysis identified 42 distinct domain types in 64 *L. pneumophila* effectors causing toxicity in the yeast model system". There is zero description in the results section of the experimental system used to express the genes in yeast in this part. Please provide this description for better understanding. Alternatively remove this sentence or mention that it will be described later in the manuscript.

We have removed this sentence from the manuscript.

2. Table 1 headers and cells are trimmed and cannot be fully read. It would be best not to convert the table into a PDF but to keep it as a spreadsheet, or at least make sure that it is fully readable before submission.

We thank the reviewer for pointing out this mishap. Table 1 is now presented as an Excel spreadsheet file.

3. "The toxicity of this effector to yeast was linked to its cysteine protease domain and was shown to be alleviated by Cys280Ser" - the figure 9A shows C280A.

To clarify, the Cys280Ser mutation was previously described as part of the set to alleviate the yeast toxicity of Lpg1290 effector, which contains a cysteine protease domain. We cite this study in the manuscript however to be consistent in our mutational analysis and expand on previous study we introduced the Cys280Ala mutation into this effector, which also completely abrogates the Lpg1290's toxicity in yeast cells, thus, in line with proposed essential role of this amino acid in catalysis. Therefore, the C280A caption, shown in Figure 9A (now part of Figure 2) is correct. We have added a descriptor in this figure's caption to clarify this point.

Reviewer #2:

SUMMARY

This study seeks to broaden understanding of the structure and function of *Legionella pneumophila* effectors proteins. With a hefty number of 368 previously known effectors, identifying and characterizing structural domains presents both a challenge and an opportunity. Through an approach that combined existing structural data, AI-built structural models and yeast functional assays, the work detailed in this manuscript provides a comprehensive, careful analysis of predicted functional domains among effectors. A number of new and important insights emerged from this work, including:

- Revealing cysteine proteases as the most prevalent, yet functionally diverse class of *Legionella* effectors. Notably, analyses more than doubled the number of cysteine proteases known prior to this work.
- Expanding the number of effectors that harbor tandem repeat motifs typically found in eukaryotic organisms and known to be functionally versatile.
- The observation that IDRs are more abundant among effectors than the rest of the

proteome.

- Predicting new functional domains including coordinates of their catalytic motifs. These are valuable data that can inform future in-depth studies.
- Revealing the prevalence of effectors harboring transmembrane helices.
- The identification of numerous cryptic domains, some predicted to have functions that have been associated with *Legionella* effectors and others predicted to have new functions.
- Identifying a subset of effectors with new, unique folds, some being predicted to share functions with effectors from other species. Probing the functionality of these unique folds through yeast growth assays which in three of the four tested effectors led to identification of critical residues. Even finding that many other domains with predicted novel folds were not lethal is useful information for future studies.
- Creating a publicly available interactive web-based database, providing immediate and unrestricted access to new, valuable data that can inform future functional studies.

We thank reviewer 2 for taking the time to critically evaluate this study. We appreciate the positive remarks of the manuscript. We are optimistic that we have addressed your questions in full below.

MAJOR POINTS

None

MINOR POINTS

- For those effectors that harbor IDRs, is there an indication of whether these regions tend to be closer to the N- or C-terminus of the protein, or elsewhere in the protein.

Interestingly, there indeed seems to be a statistically significant difference between the effectors vs. non-effectors as to the position of the IDRs. 61 of the effectors have the IDRs on the C-terminal, 17 on the N-terminal vs. 62 / 44 for the rest of proteins with the IDRs (this also illustrates the prevalence of IDRs in the effectors, since the latter group is 10x larger). This is a statistically significant difference (p.value of 0.0068). Potentially, a future study will be taking a closer look at the evolutionary importance of incorporating IDRs in effectors translocated by *L. pneumophila* and other members of the *Legionella* genus.

- In terms of the distribution of residues across the 368 effectors, are cysteines more prevalent in effector proteins than the rest of the proteome?

There seems to be no statistically significant difference in cysteine prevalence between the effectors and non-effectors. We assume that this is because the *Legionella* effectors are secreted to the host cytoplasm, not to the extracellular space.

- In many protozoa and organisms that harbor cysteine proteases, their activity can be

regulated by cysteine protease inhibitors such as cystatins. Is there an indication of whether the Legionella effectorome (or genome) harbors proteins that function as cysteine protease inhibitors?

No, the effectorome of *L. pneumophila* seems not to contain any cysteine protease inhibitors. Possible explanation for this is that as the effectors are secreted into the host cell cytosol and typically contain protein-protein binding domains that define its specificity against a host target, therefore Legionella is under no evolutionary pressure to limit their activity with cysteine protease inhibitors. An example of such regulation is for instance LegA7 protein, where C-terminal ankyrin domain repeats modulate the function of the catalytic domain.

CONCLUSION

Overall, this is a very well-written manuscript describing an elegant, thorough, and timely study. It was a pleasure to read. The clever use of advanced computational tools and traditional methods resulted in a significant expansion and refinement of current knowledge of the structure and function of Legionella effectors. In my opinion, these contributions are more than sufficient to warrant publication of this manuscript as is. The wealth of new information made available here will positively impact future studies in Legionella pathogenesis and more broadly in the host-microbe interaction field.

Reviewer #3:

The study by Patel et al., discovers a large number of functional domains and unique domains in an array of legionella effectors. They also go onto identify catalytic residues in a number of these domains and show how they affect yeast toxicity (a common functional assay for legionella effectors). This is a great resource for legionella effector community in general and possibly also for researchers exploring effectors of other pathogenic bacteria as many of these newly identified domains seem to be also present in other pathogens. My specific comments are listed below.

We thank the reviewer for praising this study as the great resource for the *Legionella* effector community. This was indeed our goal, and we hope that we have addresses all the comments the reviewer had in full.

Major comment:

1) The paper reads more like a book with too many names and results presented as a matter of fact. It would be nice for eg. to briefly mention if an effector's role in intracellular replication has been studied previously or not for each effector discussed. Or this information can also be in a table. Without this context, much of the manuscript reads like a series of observations with no biological implications immediately apparent.

We have tried to the best of our abilities to cite and mention the previous studies elucidating the role of specific effectors throughout our manuscript. At the beginning of most sections dedicated to the analysis of specific predicted functional domains we have provided a brief introduction containing references to previous studies including

any available biochemical analysis and identification, where applicable, of specific host targets. Furthermore, for effectors the models of which we have analysed in detail, we provided where available previously defined molecular data. For example, please see the description of Lpg2953; the member of the Lot DUB effector family that nevertheless appeared to lack the residues typical of this catalytic activity or of Lpg1949; the potential member of YopJ like effector group. We also included the links mentioning specific effectors on google scholar database as a column in supplemental table 1.

2) It would help the papers impact if the authors can verify at least a couple of biochemical activities predicted by AF. I understand the lack of information on substrates is a roadblock but some enzymes (eg. kinases) self-modify so this can be used to verify their activity relatively easily. Although yeast toxicity experiments are informative, biochemical assays are required to validate the domain predictions of AF.

While we appreciate reviewer's suggestion, we believe that performing such biochemical experiments would be out of the scope of this manuscript. The goal of described study is to define the repertoire of *L. pneumophila* effector domains for the bacterial pathogenesis community with in-depth structural analysis that has been validated using the yeast toxicity model – a common phenotype indicative of the biochemical function of effectors. However, defining of specific biochemical activity of effectors will need to be based on the knowledge of host targets and other molecular environment elements and thus need to be tailored for each individual effector. In this regard, performing a generic biochemical assay for only one or two functional domain groups, because of the lack of standardized assays for most of defined domain categories, would divert from the focus of the study. Furthermore, generic assays may be misleading in some cases, especially, with *L. pneumophila* effectors, where the divergence of biochemical function from the commonly described has been reported on several occasions. For example, in the case of SidJ, which has a molecular fold of a kinase but relies on Calmodulin to activate it and added polyglutamate modifications on members of the SidE family. Also, any generic biochemical assay will require the purification of recombinant effector, which cannot be done in mass and will require empirical optimisation. With this rational in mind, we would like to limit our functional assays to yeast toxicity as it provides a solid phenotype and clear path for future analysis. Described mutational analysis of predicted functionally important residues provides a clear confirmation for such prediction.

Minor comments.

In the first section of results, authors write that "This level of confidence was shown to correspond to 100% correct fold assignments in numerous independent experiments, including those performed by our groups." It would be nice to support such statements with appropriate references.

We have transferred this to Figure caption 1 and have also provided a reference, please refer to lines 1195-1196.

In the server link, overall, the database is nicely organized. But I found that when I tried to download some PDB files, the server just did not respond. Can the authors look into this? I randomly looked up a few effectors and Lpg2153 and Lpg0376 AF PDBs were not available.

We have fixed this problem in the updated manuscript

SidJ should be mentioned and discussed in the section about LP kinases.

The SidJ description is now added to the section discussing kinase domains (see lines 398-404).

Figure 1: It would be easier if the authors can include what each color means within the figure itself. Also authors should mention the specifics of what they mean by remote homology and close homology in the figure legend.

We have made the appropriate adjustments to Figure 1 and the caption.

In the "Cryptic domain activity..." section, at the very end, the authors talk about the Lpg2523/Lem26 PDE mutant still showing toxicity in yeast.

I was a little confused by the term cryptic domains. This usually means that the domain's function is unknown. This term "cryptic" is more appropriate for "unique" domains. Instead of cryptic, the authors should use the term "predicted functional domain" especially in the title of the section. In fact, authors used this term later in the description to refer to these exact domains.

In compliance with the reviewer's suggestion, we have replaced the term "cryptic" domain with "predicted functional domain" in the text. In line with this, we also have changed our nomenclature from "unique" to "cryptic" for description of domains with novel folds.

In the final section of results, the authors use ESM Fold instead of AF2. I am curious as to how the model accuracy generated using ESM fold can be compared to the AF2 model accuracy to judge which is better. Can the authors comment if they observe similar catalytic residues using both the predicted models?

AlphaFold is based on the analysis of patterns in multiple sequence alignments (MSA), while ESM is based on a large protein language model. As a result, AlphaFold excels at structural predictions for proteins with many homologs and, as a result, rich and diverse MSA. On the other hand, ESM algorithm, while on average proving less accurate models, does not depend on the MSA analysis and excels at predictions for orphan proteins (proteins with no known homologs). We have used ESM only in the cases where AlphaFold failed to provide models with pLDDT >70. AF pLDDT (predicted local distance difference test) is a measure of confidence in the model quality) and analysis shown that the diversity of the MSA for this protein is low. We have expanded the appropriate Methods section to explain this.

Despite low confidence, the AF models are structurally similar to ESM models, even that usually less compact. In all cases when we used ESM for modeling, the putative active site residues were conserved in both models.

In the final section of discussion, authors state "To conclude, by combining advanced 3D structural predictions of all reported *L.pneumophila* effectors with functional assays, we have converged on a full catalog of predicted functions and mechanisms of how this pathogen takes control and thrives inside eukaryotic phagocytic cells." I totally agree with the first part of this sentence but the manuscript does not shed light on how pathogen takes control and thrives inside phagocytic cells. This needs to be removed or reworded to accurately reflect the findings.

To address the reviewer's comment, we have removed the second part of this statement and reworded the remainder to reflect our findings more accurately (see lines 932-937).

Table 1: the table legend is not detailed enough to understand all the details of the table contents. Each column heading should be explained in the legend.

We have edited the legend to Table 1 according to reviewer's comment.

In the methods, 3D modeling section, authors seem to have split sequences for AF prediction for select effectors. These need to be mentioned in the section.

We have now mentioned the effectors of which the sequences were parsed to comply with the Alphafold2 server limit. Please see lines 950-961.

27th Sep 2024

Manuscript Number: MSB-2024-12436R

Title: Global atlas of predicted functional domains in *Legionella pneumophila* Dot/Icm translocated effectors

Dear Dr. Savchenko,

Thank you for the submission of your revised manuscript to Molecular Systems Biology. We have now received the enclosed reports from the referees that were asked to re-assess it. As you will see the reviewers are now globally supportive and I am pleased to inform you that we will be able to accept your manuscript pending the following final amendments:

1) Please check the "Author Checklist" carefully and complete all boxes - currently the general information about author, journal, and manuscript number is missing. In addition, you have indicated in the section on antibodies that it was not applicable to the study, however we note that antibodies have been used for western blots. Please update this question in the checklist. Finally, the cell lines section does not include yeast - you may mark this as not applicable if no other cell lines have been used.
2) Author contributions: Please remove it from the manuscript and specify author contributions in our submission system. CRediT has replaced the traditional author contributions section because it offers a systematic machine-readable author contributions format that allows for more effective research assessment. You are encouraged to use the free text boxes beneath each contributing author's name to add specific details on the author's contribution. More information is available in our guide to authors:

<https://www.embopress.org/page/journal/17574684/authorguide#authorshipguidelines>

3) In the Methods, please take care of the following:

- Primers: please ensure primers used are included in the Methods (or if included in table format, that the table is included in the Appendix). Currently there is a statement in the Methods that primers can be found in Supplementary Tables 5 and 6, but these tables were not uploaded to the submission.

- Please ensure that a statement on whether or not blinding was done is included in the Methods even if no blinding was done.

4) All Materials and Methods need to be described in the main text using our 'Structured Methods' format. According to this format, the Methods section includes a Reagents and Tools Table (listing key reagents, experimental models, software and relevant equipment and including their sources and relevant identifiers) followed by a Methods and Protocols section describing the methods, ideally using a step-by-step protocol format. The aim is to facilitate adoption of the methodologies across labs. Please download and fill our Reagents and Tools Table template (.docx), which you can find in our author guidelines:

<https://www.embopress.org/doi/10.15252/msb.20178071>. "

5) Please place individual sections of the manuscript in the following order: Title page - Abstract & Keywords - Introduction - Results - Discussion - Methods - Data Availability - Acknowledgements - Disclosure and Competing Interests Statement - References - Figure Legends - Expanded View Figure Legends.

6) Please move the Keywords to after the Abstract.

7) For the figures and figure legends, please take care of the following:

- Please make sure to update the callouts of all figures in the main manuscript text. All Figure panels should be called out in in number and also alphabetical order: currently Fig. 3B is called out before Fig. 2; callout for Fig. 2B before 2A; Fig. 4 called out before Fig. 3; In addition, there are some missing panels/tables: there is a callout for Fig. 1C, but Fig. 1 doesn't have any panels; there is a callout for Fig. 2E, but no such panel in Fig. 2; there are callouts for Supplemental Table 3, 4 and 5 but those tables are not uploaded

8) Tables: Table 1 and Supplementary Table 1 should be renamed to Dataset EV1-EV2. Please upload these tables as one .xsl file per table and will need its legend removed from the manuscript and added to the corresponding file in a separate tab. Please be sure to update their callouts in manuscript text.

9) Appendix file: Please upload the Appendix as a single PDF. All Figures S1-S12 should be compiled in one Appendix PDF with the nomenclature Appendix Figure S1-S12 and appropriate callouts in the manuscript text; legends for figures S1-S12 should be removed from the main manuscript file and placed below the corresponding figures in the Appendix PDF. The file should also start with a table of contents with page numbers on the title page (Appendix for Global atlas of predicted functional domains in *Legionella pneumophila* Dot/Icm translocated effectors)

10) Funding: Please ensure that all funding sources are entered into the manuscript submission system.

11) Synopsis:

- Synopsis image: The current graphic provided does not fit into our dimensions. While we can resize it for you, the graphic is too high when resized to 550 pixels wide. Please upload the image as a high-resolution jpeg file 550 pixels wide x (250-400) pixels high.

- Synopsis text: Please provide a short standfirst (maximum of 300 characters, including space), limit the bullet points to max. 5 and upload it as a separate .doc file. Please write the bullet points to summarise the key NEW findings. They should be designed to be complementary to the abstract - i.e. not repeat the same text. We encourage inclusion of key acronyms and quantitative information (maximum of 30 words / bullet point). Please use the passive voice.

12) As part of the EMBO Publications transparent editorial process initiative (see our policy here:

https://www.embopress.org/transparent-process#Review_Process), Molecular Systems Biology will publish online a Peer Review File (PRF) to accompany accepted manuscripts. This file will be published in conjunction with your paper and will include the anonymous referee reports, your point-by-point response and all pertinent correspondence relating to the manuscript. Let us know whether you agree with the publication of the PRF and as here, if you want to remove or not any figures from it prior to publication. Please note that the Authors checklist will be published at the end of the PRF.

13) Please provide a point-by-point letter INCLUDING my comments as well as the reviewer's reports and your detailed responses (as Word file).

I look forward to reading a new revised version of your manuscript as soon as possible.

Yours sincerely,

Poonam Bheda, PhD
Scientific Editor
Molecular Systems Biology

Click on the link below to submit your revised paper.

Reviewer #1:

I am happy with the revised version which improved the paper in my view. I thank the authors for responding to my initial review in detail.

Reviewer #3:

The authors have address all my concerns, the paper is an important resource and should be published.

*****Response to the Editor in Red*****

1) Please check the "Author Checklist" carefully and complete all boxes - currently the general information about author, journal, and manuscript number is missing. In addition, you have indicated in the section on antibodies that it was not applicable to the study, however we note that antibodies have been used for western blots. Please update this question in the checklist. Finally, the cell lines section does not include yeast - you may mark this as not applicable if no other cell lines have been used.

We have made the appropriate changes to the author checklist.

2) Author contributions: Please remove it from the manuscript and specify author contributions in our submission system. CRediT has replaced the traditional author contributions section because it offers a systematic machine-readable author contributions format that allows for more effective research assessment. You are encouraged to use the free text boxes beneath each contributing author's name to add specific details on the author's contribution. More information is available in our guide to authors:

<https://www.embopress.org/page/journal/17574684/authorguide#authorshipguidelines>

Completed.

3) In the Methods, please take care of the following:

- Primers: please ensure primers used are included in the Methods (or if included in table format, that the table is included in the Appendix). Currently there is a statement in the Methods that primers can be found in Supplementary Tables 5 and 6, but these tables were not uploaded to the submission.
- Please ensure that a statement on whether or not blinding was done is included in the Methods even if no blinding was done.

We have added the tables containing the list of primers used for site-directed mutagenesis in the Appendix PDF. These tables are now called "Appendix Table 3" and "Appendix Table 4". We have put a brief statement as to whether or not blinding was done in this study at the end of the methods section.

4) All Materials and Methods need to be described in the main text using our 'Structured Methods' format. According to this format, the Methods section includes a Reagents and Tools Table (listing key reagents, experimental models, software and relevant equipment and including their sources and relevant identifiers) followed by a Methods and Protocols

section describing the methods, ideally using a step-by-step protocol format. The aim is to facilitate adoption of the methodologies across labs.

Please download and fill our Reagents and Tools Table template (.docx), which you can find in our author guidelines:

<https://www.embopress.org/doi/10.15252/msb.20178071>. "

We have uploaded a table containing the reagents and tools used in this study.

5) Please place individual sections of the manuscript in the following order: Title page - Abstract & Keywords - Introduction - Results - Discussion - Methods - Data Availability - Acknowledgements - Disclosure and Competing Interests Statement - References - Figure Legends - Expanded View Figure Legends.

The manuscript is now formatted as described.

6) Please move the Keywords to after the Abstract.

We have moved the keywords.

7) For the figures and figure legends, please take care of the following:

- Please make sure to update the callouts of all figures in the main manuscript text. All Figure panels should be called out in number and also alphabetical order: currently Fig. 3B is called out before Fig. 2; callout for Fig. 2B before 2A; Fig. 4 called out before Fig. 3; In addition, there are some missing panels/tables: there is a callout for Fig. 1C, but Fig. 1 doesn't have any panels; there is a callout for Fig. 2E, but no such panel in Fig. 2; there are callouts for Supplemental Table 3, 4 and 5 but those tables are not uploaded

We have fixed the figure callouts in the main text.

8) Tables: Table 1 and Supplementary Table 1 should be renamed to Dataset EV1-EV2. Please upload these tables as one .xsl file per table and will need its legend removed from the manuscript and added to the corresponding file in a separate tab. Please be sure to update their callouts in manuscript text.

We have updated the manuscript to have Dataset EV1-2. In line with this, we have updated the table numbering accordingly in the manuscript.

9) Appendix file: Please upload the Appendix as a single PDF. All Figures S1-S12 should be compiled in one Appendix PDF with the nomenclature Appendix Figure S1-S12 and appropriate callouts in the manuscript text; legends for figures S1-S12 should be removed from the main manuscript file and placed below the corresponding figures in the Appendix PDF. The file should also start with a table of contents with page numbers on the title page (Appendix for Global atlas of predicted functional domains in Legionella pneumophila Dot/Icm translocated effectors)

We now have uploaded an Appendix PDF that contains Appendix Figures S1-12 and Appendix Tables 1-4.

10) Funding: Please ensure that all funding sources are entered into the manuscript submission system.

Done.

11) Synopsis:

- Synopsis image: The current graphic provided does not fit into our dimensions. While we can resize it for you, the graphic is too high when resized to 550 pixels wide. Please upload the image as a high-resolution jpeg file 550 pixels wide x (250-400) pixels high.

- Synopsis text: Please provide a short standfirst (maximum of 300 characters, including space), limit the bullet points to max. 5 and upload it as a separate .doc file. Please write the bullet points to summarise the key NEW findings. They should be designed to be complementary to the abstract - i.e. not repeat the same text. We encourage inclusion of key acronyms and quantitative information (maximum of 30 words / bullet point). Please use the passive voice.

Thank you for mentioning this, we have fixed the dimensions of the synopsis image, it should fit in 500 px height. We have also uploaded the synopsis text.

12) As part of the EMBO Publications transparent editorial process initiative (see our policy here: https://www.embopress.org/transparent-process#Review_Process), Molecular

Systems Biology will publish online a Peer Review File (PRF) to accompany accepted manuscripts. This file will be published in conjunction with your paper and will include the anonymous referee reports, your point-by-point response and all pertinent correspondence relating to the manuscript. Let us know whether you agree with the publication of the PRF and as here, if you want to remove or not any figures from it prior to publication. Please note that the Authors checklist will be published at the end of the PRF.

We agree to the publication of the PRF. We do not have any figures that we would like to remove.

13) Please provide a point-by-point letter INCLUDING my comments as well as the reviewer's reports and your detailed responses (as Word file).

Done.

I look forward to reading a new revised version of your manuscript as soon as possible.

Yours sincerely,

Poonam Bheda, PhD
Scientific Editor
Molecular Systems Biology

Reviewer #1:

I am happy with the revised version which improved the paper in my view. I thank the authors for responding to my initial review in detail.

We thank the reviewer for taking the time to assess the revised manuscript and providing constructive feedback. We are glad that the revisions satisfied your initial set of reviews.

Reviewer #3:

The authors have address all my concerns, the paper is an important resource and should be published.

Thank you for taking the time to read the revised manuscript. We appreciate the constructive feedback.

31st Oct 2024

Manuscript number: MSB-2024-12436RR

Title: Global atlas of predicted functional domains in Legionella pneumophila Dot/Icm translocated effectors

Dear Dr. Savchenko,

Congratulations on an excellent manuscript, I am pleased to inform you that your manuscript has been accepted for publication in Molecular Systems Biology. It has been a pleasure to work with you to get this to the acceptance stage.

Yours sincerely,

Poonam Bheda, PhD
Scientific Editor
Molecular Systems Biology
